# Representation of professions in entertainment media: Insights into frequency and sentiment trends through computational text analysis

**Sabyasachee Baruah**●*, **Krishna Somandepalli, Shrikanth Narayanan**●

Signal Analysis and Interpretation Laboratory, University of Southern California, Los Angeles, California, United States of America

* sbaruah@usc.edu

## Abstract

Societal ideas and trends dictate media narratives and cinematic depictions which in turn influence people's beliefs and perceptions of the real world. Media portrayal of individuals and social institutions related to culture, education, government, religion, and family affect their function and evolution over time as people perceive and incorporate the representations from portrayals into their everyday lives. It is important to study media depictions of social structures so that they do not propagate or reinforce negative stereotypes, or discriminate against a particular section of the society. In this work, we examine media representation of different professions and provide computational insights into their incidence, and sentiment expressed, in entertainment media content. We create a searchable taxonomy of professional groups, synsets, and titles to facilitate their retrieval from short-context speaker-agnostic text passages like movie and television (TV) show subtitles. We leverage this taxonomy and relevant natural language processing models to create a corpus of professional mentions in media content, spanning more than 136,000 IMDb titles over seven decades (1950-2017). We analyze the frequency and sentiment trends of different occupations, study the effect of media attributes such as genre, country of production, and title type on these trends, and investigate whether the incidence of professions in media subtitles correlate with their real-world employment statistics. We observe increased media mentions over time of STEM, arts, sports, and entertainment occupations in the analyzed subtitles, and a decreased frequency of manual labor jobs and military occupations. The sentiment expressed toward lawyers, police, and doctors showed increasing negative trends over time, whereas the mentions about astronauts, musicians, singers, and engineers appear more favorably. We found that genre is a good predictor of the type of professions mentioned in movies and TV shows. Professions that employ more people showed increased media frequency.

**Data Availability Statement:** The data can be found at https://github.com/sabyasachee/mica-profession.

**Funding:** The study was done at Signal Analysis and Interpretation Laboratory, University of Southern California, which is supported by a research award from the U.S. Chamber of Commerce Foundation (https://www. uschamberfoundation.org/). The funders had no role in study design, data collection and analysis, decision to publish, or preparation of the manuscript.

**Competing interests:** The authors have declared that no competing interests exist.

## Introduction

Entertainment media, available in rich variety and diverse forms ranging from traditional feature films, television shows, and theatrical plays to contemporary digital shorts and streaming content, can profoundly impact audience perceptions, beliefs, attitudes, and behavior. Media narratives aim to inform and engage us with stories about the culture, lives, and experiences of different communities of people, including reflecting societal ideas and trends. They shed light on various social, economic and political issues, educating and creating awareness on different aspects of life. Cultivation theory suggests that prolonged exposure to the content we see on TV shapes our outlook and makes us believe that to be our reality [1]. Several social studies have explored the application of cultivation theory and confirmed its validity [2–4]. However, there are social scientists that have also questioned the validity of the cultivation theory because it ignores socioeconomic factors [5], living conditions [6], and differences in the portrayal of violence on television [7]. Still, this discussion about television and the cultivation of beliefs inspires us to examine their relationship. With the increasing number of film releases (theatrical and home/mobile entertainment market increased by 4% in 2019 [8]) and the large amount of time people spend with major media (US people watched TV for around 4 hours daily [9]), media experiences influence our views and choices in many spheres of life; this includes our professional and career choices.

Professions are paid, skilled work we perform to provide services to people and earn a livelihood. They define our role in society and allows us to contribute to a nation's economy. The distribution of the country's population in different occupations provides valuable information to business leaders, policymakers, educational institutions, students, and job seekers, to understand the labor market and make decisions. Therefore, it is critical to study and assess the changes in the occupational structure of nations. One of the primary factors that affects the professional distribution of a country is the career choices its inhabitants make regarding what educational and professional pathways to pursue. Personal interests, family expectations, contemporary culture, and media exposure influence their choices [10, 11]. Of particular interest to us in this work is the relationship between people's career choices and the media representation and portrayal of professions. Several prior works have studied this connection. Undergraduate students have indicated that the portrayal of the advertising industry in two popular TV shows—*Mad Men* and *Trust me*, prompted them to enroll in advertising courses [12]. Majoring in journalism could be predicted by the student's media and technology use [10]. US Navy recruitment went up by 500% after the release of the movie *Top Gun* because many young men were inspired by the character *Pete "Maverick" Mitchell*, a US naval aviator, played by Tom Cruise [13]. Similarly, the detective character *Dana Scully*, of the TV series *The X-Files*, played by actor Gillian Anderson, inspired many young women to pursue a career in STEM [14]. A survey of 1005 currently employed people in the US found that 58% of them attributed their career inspiration to some book, TV show, movie, podcast, or video game [15]. Therefore, the portrayal of professions in media plays a significant role in our career decisions, affecting the occupational distribution of society.

Several studies have examined the nature of the portrayal of different professions in popular media, such as lawyers [16], accountants [17], physicians [18], and police [19]. While these past studies closely examined the personality of the character portrayed in the profession of interest, their methods are not scalable because the authors manually viewed the movie or read the transcripts to infer the character's personality. Such studies can not examine more than a few hundred movies or TV shows at a time. The set of personality attributes also varied between different works. Therefore, there is a need to conduct such media studies of professions more systematically and computationally.

Our objective is to conduct a computational study of the representation of professions in media content, spanning a large set of movies and TV shows over a time period. We rely on textual data from media subtitles for this study. Specifically in this work, we use *professional mentions* as a proxy for the representation of professions in movies. Professional mentions are job titles (doctor, engineer, cop, lawyer, etc.) used to indicate a profession within an utterance. Word mentions have been previously used to study trends of different societal functions, for example, education, culture, and language use patterns [20–22]. Michel et al. introduced the Google Books corpus, which contains digitized copies of more than five million books [23]. They used n-gram frequencies to track the size of the English lexicon, word usage, grammatical structures, popularity index of individuals, etc., over time. Brysbaert et al. argued that word frequency measures of media content were better than those calculated from written sources for psycholinguistic research [24]. Inspired by these works, we use mentions of profession related words to study the representation of professions in media content. The following are the contributions of our work:

1. We develop a scalable and searchable taxonomy of professional titles, professional Word-Net synsets and occupational groups of the Standard Occupational Classification (SOC) system [25].

2. We describe and share a new corpus of professional mentions, spanning 4,000 professions, 136,000 movies and TV shows, ranging over the years 1950 to 2017 (almost 7 decades) created by analyzing job title occurrences in media subtitles [26].

3. We study the relationship between real-world employment trends and mentions of different occupational groups in media content.

4. We analyze the frequency and contextual sentiment trend of professional mentions in media over time [27, 28]. We also investigate the correlation between incidence of professional mentions and the genre, title type, and country of production of the movie or TV show.

The rest of the paper is organized as follows: the Related Work section reviews some past studies about media representation of professions. It also gives the technical background of the computational models and knowledge bases we use in our work. The Data section describes the profession gazetteers and the subtitles dataset we use to search professional mentions in media content. The Methodology section is divided into two parts: Taxonomy Creation and Profession Search. The first subsection explains how we create a searchable taxonomy of job titles. The second subsection describes how we use this taxonomy to search and annotate professional mentions in media subtitles. The Analysis section analyzes the media frequency and sentiment trends of different professions, and their correlation with the real-world employment figures. We conclude with a discussion of the results of our analysis and opportunities for further research.

## Related work

We summarize some past studies on media representation of professions. These studies have typically examined a small set of movies, usually manually, and their methods cannot be scaled easily to other occupations. We build upon various well-established natural language processing (NLP) methods and lexical resources to address these limitations. We leverage job title gazetteers and WordNet synsets to create a searchable taxonomy of professions. We prune non-professional mentions of job titles using word sense disambiguation and named entity

recognition, and find the targeted sentiment of the remaining mentions. We briefly explain the theoretical background of each method and highlight relevant work.

## Social scientific studies

Several past works have studied the portrayal of professions in entertainment media. Asimow studied the representation of lawyers in 284 films and found most portrayals to be negative [16]. Dimnik et al. examined the representation of accountants in 121 movies and extracted six character stereotypes of the accountant personality [17]. Flores investigated physicians' image depictions in 131 films and found that they were mostly depicted as greedy and uncaring [18]. Pautz used a sample of 34 films containing more than 200 police (cop) characters and found that most cops were shown as good, hard-working, and competent law-enforcement officers [19]. Kalisch et al. analyzed 670 nurse and 466 physician characters in novels, movies and television series, and concluded that compared to physicians, nurses portrayed in the media were consistently less central to the plot, less intelligent, less rational, and less likely to exercise clinical judgement [29]. Smith et al. investigated gender representation of occupations in films, prime-time programs, and children TV shows, and found that women are grossly underrepresented compared to men in science, technology, engineering and math jobs (STEM) [30]. These works involved extensive human coding and profession-specific analysis which is not reproducible on a large scale. The present study offers a complementary view in the sense it trades off a smaller scale character-centric study for a larger scale lexical analysis, focusing on character utterances instead of personality traits. The remaining sections describe the computational methods used to achieve the same.

## Named entity recognition

Named entity recognition (NER) is a classic NLP task of finding entity mentions in text and classifying their type. Traditional NER primarily targets person, organization, and location entity types. Most NER datasets only contain labels for these three types of entities [31]. The OntoNotes 5 dataset increased the number of entity types to 18 by including nationalities, products, events, and numeric values [32]. However, it does not contain any professional titles. Fine-grained NER extends traditional entity recognition by expanding the entity set to include hundreds of named categories. Ling et al. created a benchmark dataset for fine-grained NER that labels 112 different entity types but it only contains a few professional titles [33]. Sekine built an ontology for named entities and defined professional titles as vocational attributes for the person-entity type [34]. Mai et al. used this entity hierarchy to annotate English and Japanese sentences and evaluated different fine-grained NER models [35]. However, they did not include the professional attributes in their labeling set. The Text Analysis Conference Knowledge Base Population (TAC KBP) track of entity discovery and linking introduced job titles as entity types to model the *person:title* relationship [36]. The Stanford CoreNLP NER model used the KBP 2017 dataset to create regular expression-based rules for finding professional titles in text [37]. However, we observed that it missed many of the professional mentions in media text.

In the absence of labeled data, entity gazetteers are often used to find candidate spans for named entities. Gazetteers are curated lists of entities that improve NER performance when combined with supervised models. Lin et al. used gazetteers to identify text subsequences for the region-based encoder and improved the state-of-the-art NER performance on the ACE2005 benchmark [38]. Liu et al. combined dictionary lookups with semi-Markov CRF (conditional random field) architectures and achieved comparable results with more complex neural models on the CoNLL 2003 and OntoNotes 5 datasets [39]. Several gazetteers of job

titles are available (Gate job titles, fluquid). Government and international organizations use some standard gazetteers of job titles to collect occupational data. For example, the International Labor Organization uses the International Standard Classification of Occupations [40], and the US Bureau of Labor Statistics maintains the Standard Occupational Classification (SOC) system [25]. Aside from careful human coding, such gazetteers can be constructed using different automated methods, including those that leverage existing knowledge bases, such as Wikipedia and WordNet [41, 42]. In this work, we use the SOC taxonomy to get the initial list of job titles and professional groups.

## WordNet

WordNet is a widely-used lexical resource in English [43]. It groups words into synonym sets called synsets. Synonymy is the semantic relationship between words of similar meaning. Polysemous words have multiple meanings and belong to more than one synset. For example, the word "conductor" is present in three synsets—*conductor.n.01* (the person who leads a musical group), *conductor.n.02* (a substance that readily conducts electricity and heat) and *conductor. n.03* (the person who collects fares on a public conveyance). The name of a synset is composed of three dot-separated literals. The first literal is the lemmatized form of the main word of the synset, the second literal is the part-of-speech, and the third literal is the sense index. Synsets are tagged by semantic classes [44]. Synset *A* is a hyponym of synset *B* if it is a more specific form of *B*. For example, *allergist.n.01* (doctor specialized in the treatment of allergies), *surgeon. n.01* (doctor specialized in surgery) and *veterinarian.n.01* (doctor practicing veterinary medicine) are all hyponyms of the more general synset, *doctor.n.01* (a licensed medical practitioner).

WordNet has been used to construct entity gazetteers. Toral et al. leveraged WordNet's noun hierarchy to build person and location gazetteers [45]. Maginini et al. used WordNet to identify trigger words and gazetteer terms for English NER [46]. Boteanu et al. expanded a shopping taxonomy for efficient product search by matching the product names to WordNet synsets [47]. In this work, we use WordNet synsets to extend the SOC taxonomy and create a searchable dictionary of professions. WordNet synsets are also the target labels for the word sense disambiguation task, an integral part of our search pipeline.

## Word sense disambiguation

Word sense disambiguation (WSD) is the task of assigning words in context to their most appropriate sense. Wordnet usually serves as the sense inventory that provides the target senses. The same word can express different meanings depending upon its context. Consider the word "conductor" in the following two sentences—"Conductors communicate with the musicians through hand gestures" and "Metals are good heat conductors". The former refers to a person directing the music of an orchestra, denoted by the synset *conductor.n.01*. The latter means a heat-conducting substance, denoted by the synset *conductor.n.02* (See Sec WordNet). Many NLP tasks such as machine translation, information retrieval and question answering use WSD in their text-processing pipeline [48–51].

Knowledge-based and supervised approaches have tackled WSD, with the latter usually outperforming the former. Raganato et al. standardized the evaluation framework for WSD and used a combination of SenseEval [52, 53] and SemEval [54–56] datasets to compare the performance of different WSD models [57]. Raganato et al. treated WSD as a sequence labeling task and used an LSTM with attention layers to find the sense of all sentence words jointly [58]. Huang et al. constructed context-gloss pairs and converted WSD into a sentence pair classification task [59]. Kumar et al. produced gloss embeddings using the WordNet graph and

combined them with contextual vectors to find the word sense [60]. Bevilacqua et al. extended this method by adding hypernym and hyponym relational knowledge to construct the synset vectors and achieved state-of-the-art WSD performance [61]. In this work, we use WSD to remove non-professional mentions of job titles in media subtitles.

## Sentiment analysis

Sentiment analysis or opinion mining is the task of finding the sentiments, opinions, attitudes, appraisals and emotions towards entities or their attributes expressed or implied in text [62]. Sentiment is always targeted at some entity or towards some attribute of the entity. The target entity or attribute can be a person, organization, issue, product, service, topic or event. Such target-oriented opinion mining is called aspect-based sentiment analysis (ABSA).

We use professional mentions as opinion targets for sentiment analysis to find how positively or negatively different professions are talked about in media stories. The task of profession ABSA is to find the sentiment orientation of the opinion expressed towards the person or group of persons referred to by their job title. If the job title does not refer to any profession, or people employed in the profession, the sentiment is deemed neutral. The following example sentences show the sentiment label of the job title word, marked in bold.

1. *Harry Floyd was a great **actor**.* (POSITIVE)
   Explanation: Actor refers to Harry Floyd who is described as great.

2. *But that damn **vet** kept ordering test after test after test!* (NEGATIVE)
   Explanation: The speaker uses a swear term, *damn*, to address the veterinarian and criticizes his or her action.

3. *Fine, then we get the armor and reverse **engineer** it.* (NEUTRAL)
   Explanation: The word *engineer* refers to an action, not a profession.

4. *You're going to be a lousy **architect**.* (NEUTRAL)
   Explanation: The person towards which the negative sentiment is expressed, is not yet an architect.

Benchmark ABSA datasets exist for several domains such as question answering forums, customer reviews and tweets [28, 63, 64]. Dong et al. proposed an adaptive recursive neural network for target-dependent twitter sentiment classification, that propagated the sentiments of words to their targets depending upon the context and syntactic relations [28]. Tang et al. used LSTMs to model the left and right context of the target entity for twitter sentiment classification [65]. Memory networks and graph convolutional neural models have also been proposed [66, 67]. Recently, several transformer networks have been introduced for the ABSA task and have achieved state-of-the-art performance [68]. Sun et al. constructed sentences containing the aspect expression and the sentiment orientation, and fine-tuned a pre-trained BERT [69] model for the ABSA task [70]. Xu et al. trained a BERT model jointly for reading comprehension and ABSA, and showed performance improvement on both tasks [71]. Zeng et al. proposed dynamically weighing or masking the attention weights of the sentence words depending upon its distance from the aspect expression [27]. In this work, we use ABSA models to find the sentiment expressed toward professions in media content.

## Data

We search for mentions of job titles in entertainment media content to study the representation of professions. We use the Standard Occupational Classification (SOC) system [25] to create a searchable taxonomy of professional titles. We apply this taxonomy to find professional

mentions in media (movie) subtitles, for which we use the OpenSubtitles corpus [72]. This section describes the SOC taxonomy and the OpenSubtitles dataset.

## Standard Occupational Classification taxonomy

The Standard Occupational Classification (SOC) system is a profession taxonomy, maintained by the US Bureau of Labor Statistics [25]. It arranges professions in four tiers: major, minor, broad, and detailed. The detailed tier contains a set of professions, closely related by work. Fig 1 lists all 23 major SOC groups, and shows a portion of the taxonomy's subtree rooted at *Management Occupations* major SOC group. As shown in the figure, the profession *Governor* occurs in the following SOC hierarchy: *Management Occupations* (major) → *Top Executives* (minor) → *Chief Executives* (broad) → *Chief Executives* (detailed) → *Governor* (profession). The SOC taxonomy contains 6520 unique professions.

## OpenSubtitles

We used the English subset of the OpenSubtitles dataset [72], which contains 135,998 subtitle files corresponding to a variety of media content from the years 1950 to 2017. Each subtitle file is mapped to a unique IMDb title. More than 94% of the IMDb titles are movies or TV show episodes, and the rest are made up of video games, TV shorts, TV mini-series, etc. Subtitle files for IMDb titles released before 1950 are available, but we excluded them because there were very few titles for each year (less than 100) and it would not have been a representative sample for that period's media content. The subtitle files of our dataset contain around 126 million sentences and 942 million words.

Fig 2 shows some media metadata statistics of our subtitles dataset. The first panel shows the temporal distribution of movie and TV show IMDb titles by each decade. The number of media titles increases with time, with TV episodes surpassing movie releases in the later years. The number of TV episodes is more than three times the number of movies in the most recent decade (2010-2017). The second panel shows the distribution of the top ten genres of the IMDb titles in our dataset. An IMDb title can have multiple genres. Drama and Comedy are the two most common genres, covering more than 80% of the IMDb titles. The third panel shows the distribution of the top ten most common countries where the production company is based. About 68% of the time, the production company was based in the US or the UK.

## Methodology

We create a corpus of professional mentions by searching job titles in the OpenSubtitles dataset. We use this corpus to study the relationship between media portrayal of professions and real-world employment trends. However, we cannot use the SOC taxonomy directly to find professional mentions in media content because of the following reasons:

1. Most of the SOC job titles are very specific multi-word phrases, for example, *Department Store Manager*, *Registered Occupational Therapist*, *Television News Video Editor*, etc. Such detailed words are rarely spoken in everyday conversations (including those captured in the subtitle transcripts of media considered here). They instead include simpler unigram professional words like *Manager*, *Therapist*, *Editor*, etc. Less than 7% of the SOC job titles are unigrams.

2. The mere occurrence of a job title in text does not mean it refers to some profession. For example, consider the sentence—*I made a peach **cobbler** for the party*. *Cobbler* is a job title, but here refers to a type of food. Both the lexical form and the context decides whether the word is a professional mention or not.

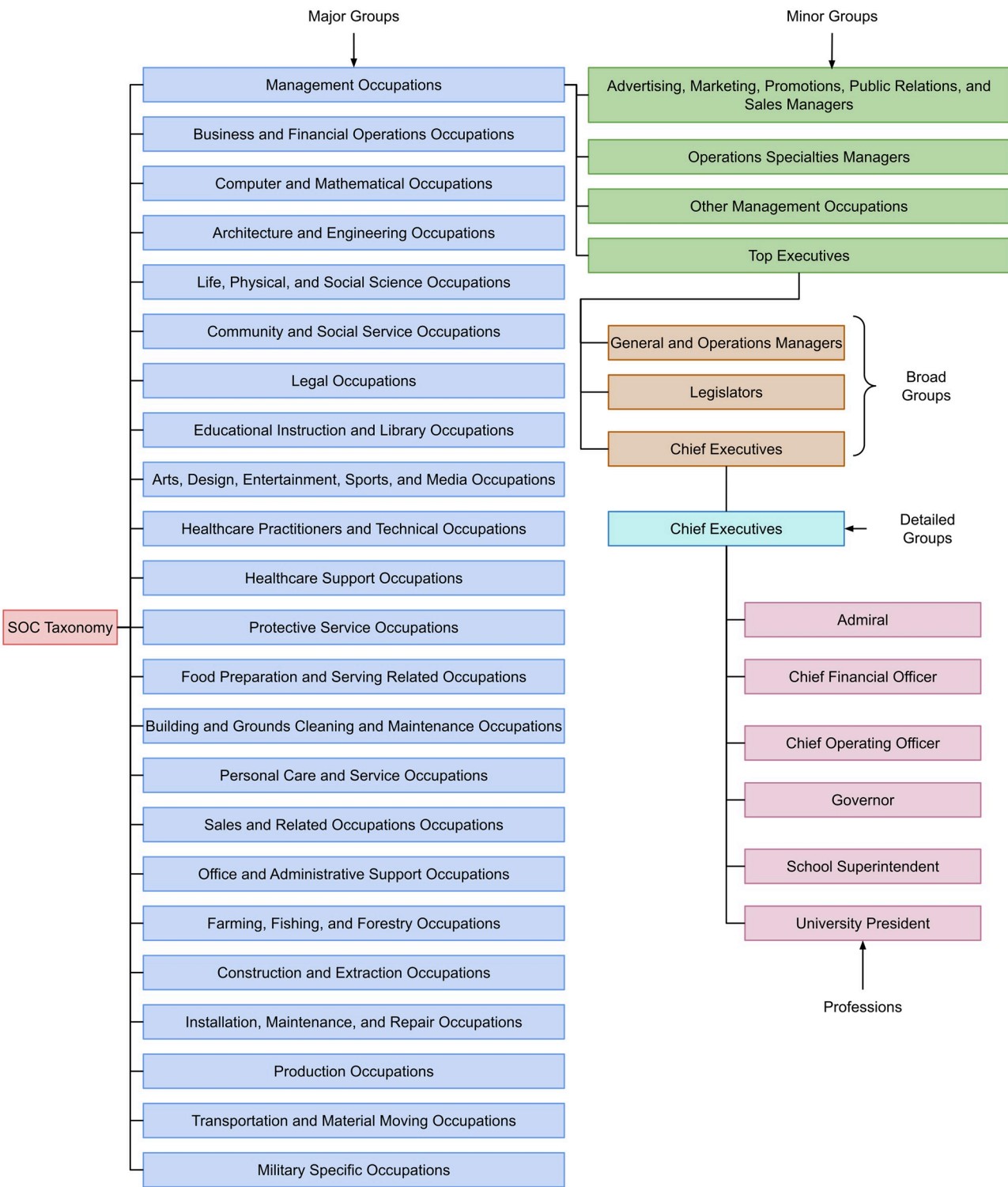

**Fig 1. Standard Occupational Classification (SOC) system.** The Standard Occupational Classification (SOC) is a 4-tiered profession taxonomy: major, minor, broad, and detailed groups. Detailed groups contain a set of closely-related professions. This figure shows the *Management Occupations* (major) → *Top Executives* (minor) → *Chief Executives* (broad) → *Chief Executives* (detailed) → *Admiral, Chief Financial Officer, . . ., University President* (professions) branch.

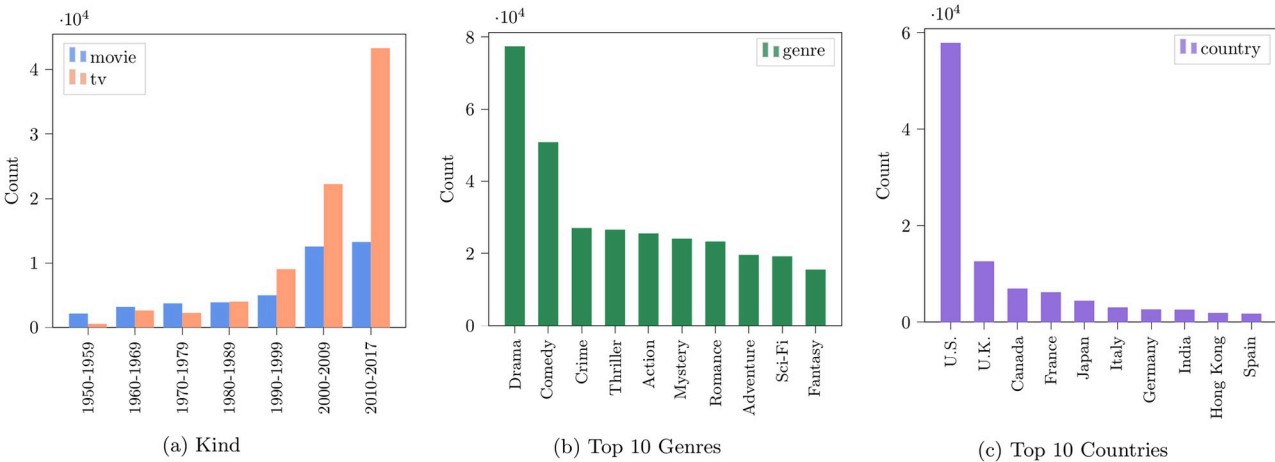

**Fig 2. Descriptive statistics of the OpenSubtitles dataset.** a) Distribution of IMDb title type (movie/TV) by year. b) Distribution of the top ten genres. c) Distribution of the top ten production countries. We use the OpenSubtitles dataset between the years 1950 and 2017.

Therefore, in order to make the SOC taxonomy searchable, we need to extend its list of job titles to include simpler, more common words, and have a disambiguation model to filter non-professional usages of job titles. Fig 3 outlines the complete pipeline of expanding the SOC taxonomy, creating the corpus of professional mentions, and analyzing its frequency and sentiment trends. The figure uses the *cobbler* profession to exemplify the corpus creation method. As shown in the figure, the Taxonomy Creation section describes how we expand the SOC system, and create a searchable profession taxonomy. The Profession Search section explains the NLP techniques we apply to find the professional mentions and its targeted sentiment.

### Taxonomy creation

We use WordNet synsets to create a searchable profession taxonomy. Fig 4 shows an example of finding new professions from the SOC job titles: *Orchestra Conductor* and *Train Conductor*.

We use the following method to expand the SOC taxonomy.

**Find Substrings**: Given a SOC job title, we split it into substrings and join them cumulatively from the end to find candidate job titles. For example, given the job title *Chief Executive Officer*, we find the new candidate titles—*Officer* and *Executive Officer*. We do not split titles that contain conjunctions, prepositions, punctuations, or those that are abbreviations (all letters are in upper case), or if they contain more than five words. In Fig 4, *Conductor* is a new job title we find from the SOC words, *Orchestra Conductor* and *Train Conductor*.

**Find WordNet synsets**: We find WordNet synsets of the candidate job titles. We retain only those synsets whose semantic class is *noun.person* or *noun.group*. We add the hyponym synsets of the retained synsets.

**Remove Non-Professional Synsets**: We manually check the list of synsets and remove those that do not refer to any profession. As shown in Fig 4, *conductor.n.02* denotes some heat or electricity conducting substance, which is not a profession and therefore, it is removed. The final curated list contains **1615** professional synsets.

**Synonym Expansion**: We collect all synonyms of the professional synsets. These are the new job titles, which we add to the SOC taxonomy. As shown in Fig 4, synsets *conductor.n.01* and *conductor.n.03* contribute the new job titles: *Music Director*, *Bandleader*, *Bandmaster* and

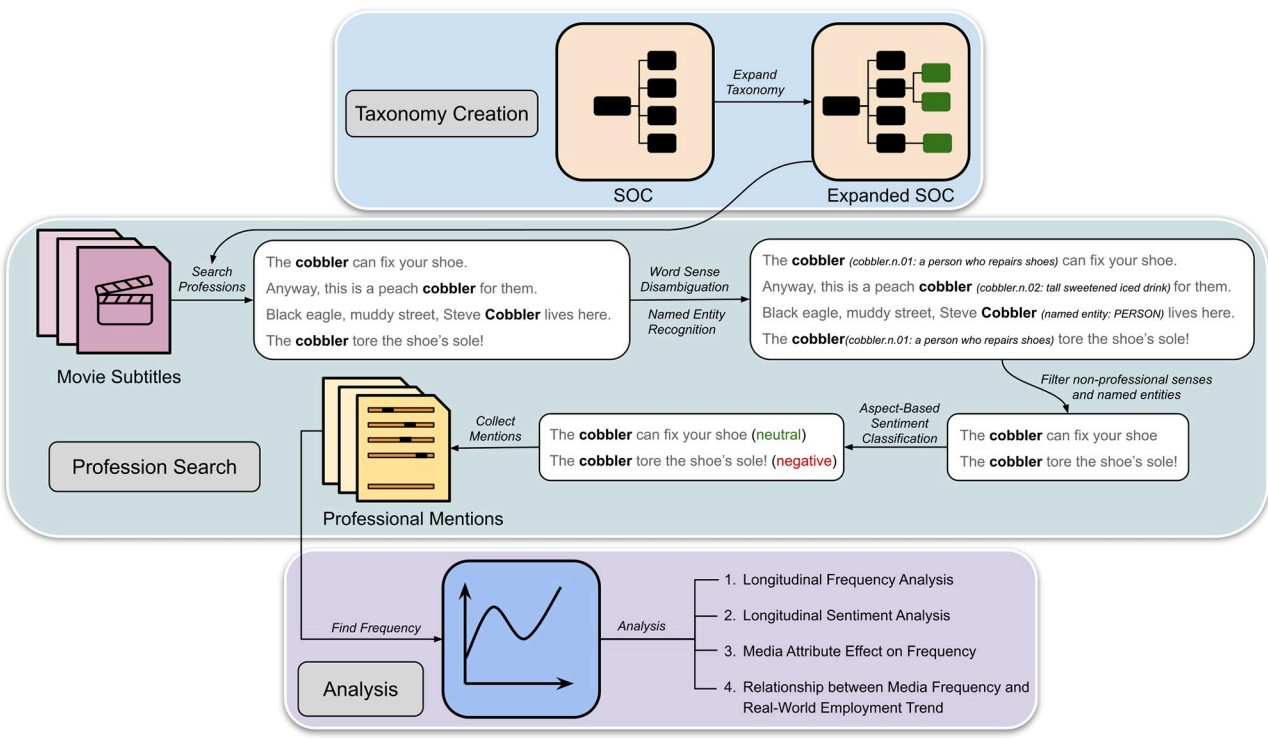

**Fig 3. Profession taxonomy creation and analysis.** The full pipeline for creating the profession taxonomy and the corpus of professional mentions, and analyzing the subtitle frequency and sentiment of professions.

*Conductress*. The final taxonomy contains **10952** job titles. The number of unigram titles increased from 426 in SOC to 1881 in the expanded taxonomy.

**SOC Mapping**: We map the job titles in the expanded taxonomy to SOC major groups. This allows us to compare the employment in different SOC professional groups with their frequency in media content. We create the mapping through a semi-automatic process. Given a job title *x*, we first find the SOC major groups that contain *x* or has some job title which contains *x* as a substring. If there exists exactly one such SOC major group, we map *x* to it. Otherwise, we examine the professional synsets of *x* and find the mapping manually. Often, different synsets of a job title map to different SOC groups. For example, from Fig 4, we observe that the two synsets of *Conductor*, *conductor.n.01* and *conductor.n.03*, map to two different SOC major

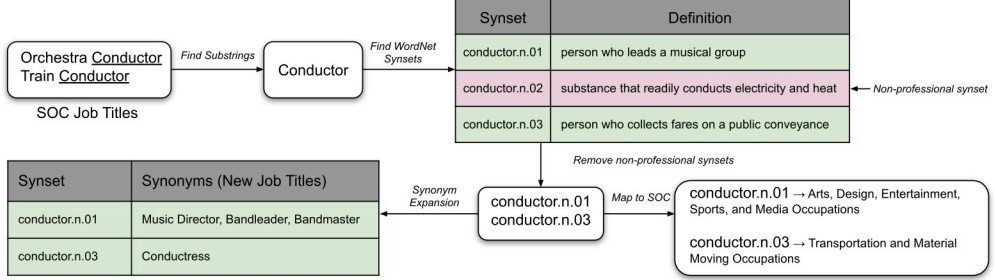

**Fig 4. Profession taxonomy creation.** We use WordNet synsets to expand the SOC taxonomy and map to SOC major groups. This figure shows how we find new job titles from the SOC job titles, *Orchestra Conductor* and *Train Conductor*.

**Table 1. Profession taxonomy sizes.**

|  | Professions | Unigram Professions | Synsets |
|---|---|---|---|
| SOC Taxonomy | 6520 | 426 | - |
| Expanded Taxonomy | 10952 | 1881 | 1615 |
| SOC-mapped Taxonomy | 500 | 409 | 562 |

We expand the SOC taxonomy to create the Expanded taxonomy. The SOC-mapped taxonomy is a subset of the Expanded taxonomy. It has been mapped to SOC major groups.

groups, depending upon their respective definitions. We only perform the mapping for the top **500** most occurring job titles in media subtitles. These job titles belong to **562** professional synsets and cover more than 94% of the professional mentions.

Table 1 shows the number of job titles and synsets in the different profession taxonomies. The SOC taxonomy does not contain any synsets and therefore, is not searchable. The expanded taxonomy adds professional synsets and quadruples the number of unigram job titles, making it searchable. The SOC-mapped taxonomy is a subset of the expanded taxonomy which we use to study the relationship of media frequency of professions with their employment trend. Fig 5 shows the structure of the SOC-mapped profession taxonomy. It contains three tiers: SOC major groups, WordNet synsets and job titles. The figure only shows five SOC major groups of the complete taxonomy.

## Profession search

We search mentions of the expanded taxonomy's job titles in the OpenSubtitles corpus. We apply NER and state-of-the-art WSD techniques to prune non-professional mentions. Finally, we train an ABSA model on sentiment-annotated subtitle sentences, and use it to tag professional mentions with their sentiment polarities.

**Mention search.**   We search the subtitle sentences to find mentions of job titles. We create a word-document search index using the Whoosh Python package [73] for quick retrieval of mentions. We also search for the plural form of the job title while finding its mentions. Paranthesized mentions, speaker references (*Referee: The match will begin shortly!*) and lyrical mentions are removed.

**Removing non-professional mentions.**   As discussed in the Methodology section, not all job title mentions refer to some profession. We remove non-professional mentions using WSD and NER methods. We apply the EWISER (Enhanced WSD Integrating Synset Embeddings and Relations) WSD model to find the mention's sense [61]. The EWISER model achieved state-of-the-art performance on the WSD benchmark dataset [57], reporting an overall F1 of 80.1. We apply the Stanford CoreNLP NER model [37] to find the named entity tags of words. We use the following rule to find professional mentions. A job title mention refers to a profession if 1) the predicted sense belongs to the set of professional WordNet synsets of the expanded taxonomy (see section Taxonomy Creation), and 2) it is not the name of an organization, or of a person who is cast in the corresponding IMDb title. We remove the non-professional mentions of job titles using the above method. The remaining mentions form our corpus of professional mentions. Fig 3 shows the steps to remove non-professional mentions for the *cobbler* job title.

To evaluate our rule-based model of finding professional mentions, we randomly sample 200 job title mentions and manually annotate their professional label. The test set contained 123 professional mentions and 77 non-professional mentions. Our model correctly predicted

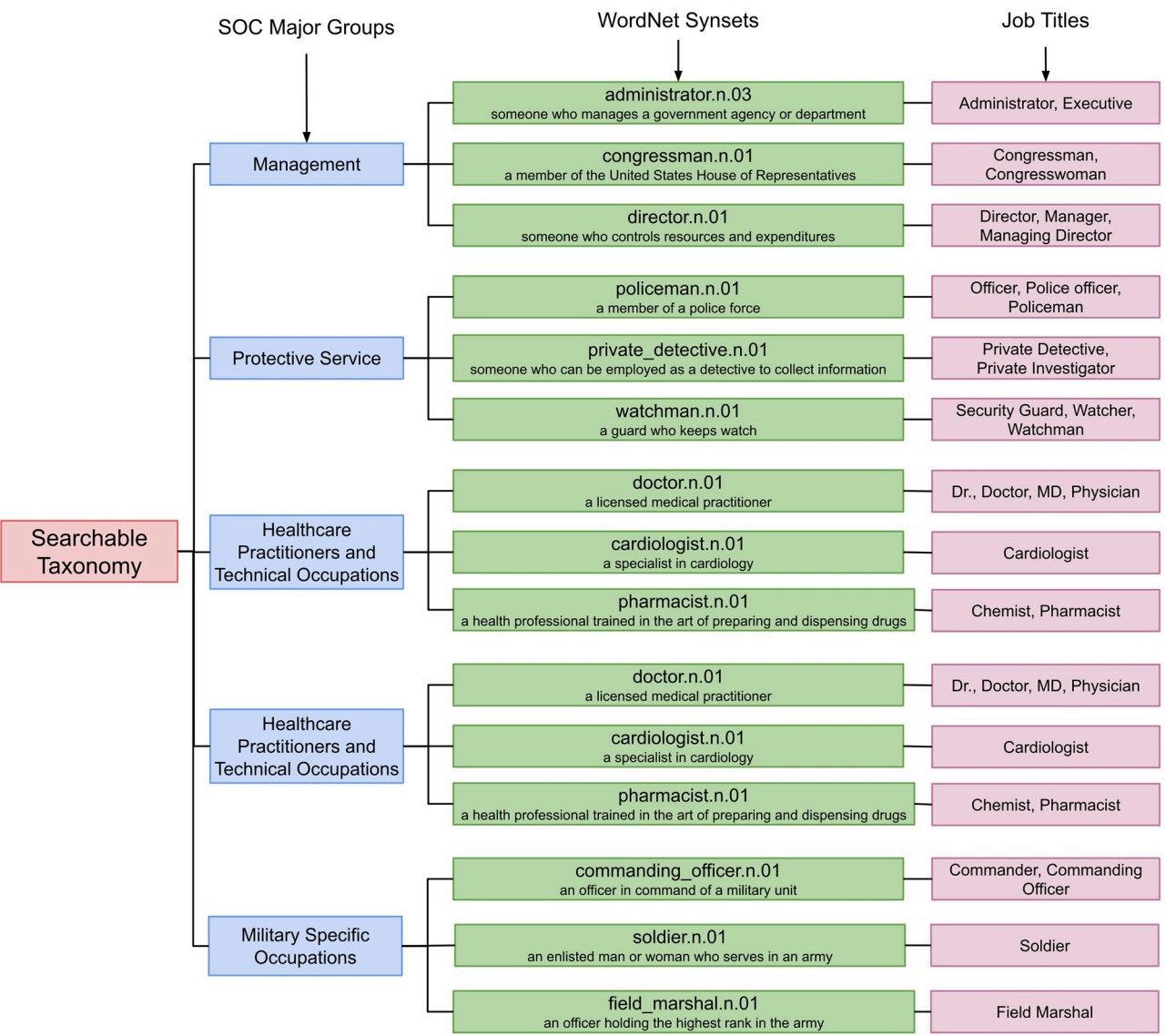

**Fig 5. Searchable profession taxonomy.** The SOC-mapped expanded profession taxonomy contains 3 tiers: SOC major groups, WordNet synsets, and job titles. The synsets and unigram job titles make the taxonomy searchable. This figure shows a few nodes of the SOC-mapped taxonomy, which contains 500 job titles and 562 synsets.

the professional label for 83.5% of the mentions, with 94.12% precision and 78.05% recall. Therefore, our corpus of professional mentions has a 5.88% false-positive rate.

**Determining expressed sentiment.** We tag each professional mention with the sentiment (positive, negative, or neutral) expressed towards them in the subtitle sentence (see section Sentiment Analysis). We apply the LCF (Local Context Focus) BERT model to find the targeted sentiment [27]. The LCF model defines a value called semantic relative distance (SRD) for each word in the sentence. SRD is the absolute difference between the word position and the target (professional mention) position in the sentence. The model masks or weighs down the output features of words whose SRD is larger than some threshold, that is, the model reduces the effect of words that are farther away from the profession word in determining the

**Table 2. Sentiment-annotated professional mentions dataset.**

|  | Positive | Negative | Neutral | Professions |
|---|---|---|---|---|
| Train | 2,431 | 1,409 | 3,915 | 85 |
| Validation | 345 | 167 | 366 | 11 |
| Test | 540 | 107 | 333 | 11 |
| Total | 3,316 | 1,683 | 4,614 | 107 |

The annotations are crowdsourced using Amazon Mechanical Turk. The train, validation and test sets do not share professions.

sentiment. The LCF model achieved state-of-the-art accuracy on the Twitter ABSA task [28], recording 75.78 F1 score. It also obtained high scores on the customer reviews dataset [64] of laptops (79.59 F1) and restaurants (81.74 F1). We train the LCF model using sentiment-annotated professional mentions.

We crowdsourced sentiment annotations using Amazon Mechanical Turk. We trained Turkers using expert-labelled examples, and then asked them to annotate the sentiment of five sentences. We selected only those annotators who correctly annotated all five sentences. 52 annotators qualified our test. These annotators then labelled the sentiment of 15,000 professional mentions: two annotations per mention. We retained only those mentions with identical annotations. We were left with 9613 sentiment-annotated professional mentions: 3,316 positive, 1,683 negative, and 4,614 neutral. The dataset contains mentions of 107 professions. To train the LCF model, we divide the dataset into train, validation and test sets. Professions, whose mentions occur in the training set, do not appear in the validation and test set. This prevents the model from overfitting to the target professions of the training set, encouraging it to learn the sentiment from the context and handle mentions of unseen professions. Table 2 shows the distribution of the sentiment classes and the number of professions in each set.

We tune the following hyperparameters of the LCF model on the validation set: *SRD*, architecture type (*CDM* or *CDW*), L2-regularization, dropout, embedding dimension, and hidden dimension. The model achieved 87.76% accuracy and 83.22 F1 on the test set. We apply the trained model to find the targeted sentiment of each professional mention of our corpus.

In total, our corpus of professional mentions contains 3,657,827 mentions, covering 4,073 professions. The corpus contains mentions from 133,133 IMDb titles, ranging between the years 1950 to 2017. The top 500 most occurring professions, which have been mapped to SOC major groups (see section Taxonomy Creation), cover more than 94% of the mentions.

## Analysis

We study profession representation in media according to the frequency of their mentions and the sentiment expressed towards them in subtitles. We also analyze the effect of media attributes like genre, location of the production company, and title type, on the incidence and sentiment of different professions. Lastly, we investigate the relationship between the trends of media frequency and the real-world employment statistics of professions.

### Profession frequency

We calculate the media frequency of a profession as the total number of professional mentions (both singular and plural form, for example, *advocate* and *advocates*) divided by the total number of *n*-grams in the subtitles. Here, *n* equals the number of words in the profession phrase, for example, *doctor* is a 1-gram, *chief executive officer* is a 3-gram, etc. We calculate the frequency of SOC major groups by adding the frequencies of professions mapped to it (see

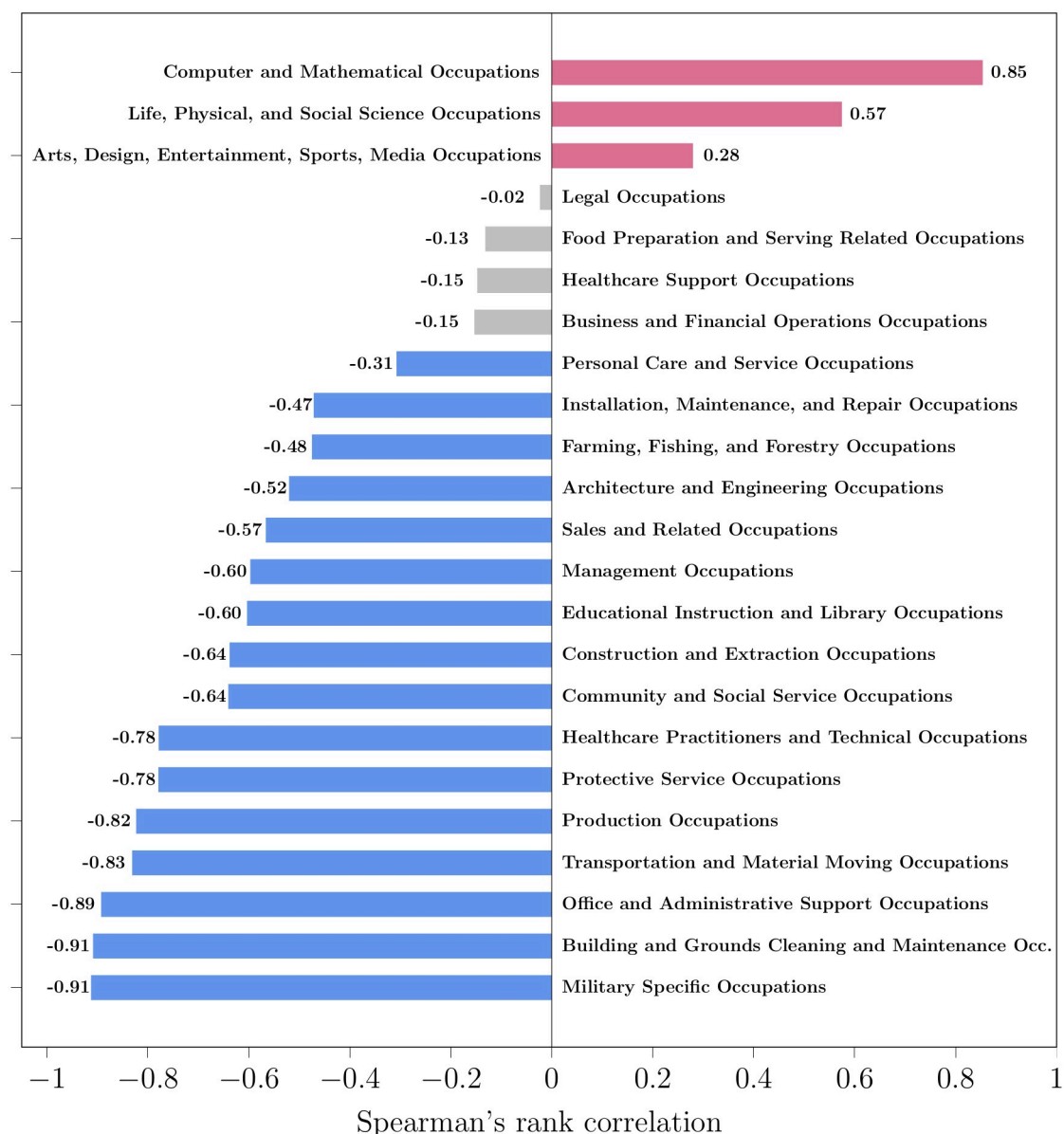

**Fig 6. Spearman's rank correlation coefficient of the media frequency of SOC groups vs time.** The red-colored bars have positive correlation (increasing trend), and the blue-colored bars have negative correlation (decreasing trend). The grey-colored bars mean that the correlation is not statistically significant ($\alpha = 0.05$).

section Taxonomy Creation). This frequency measure is motivated by the Google-ngrams study [23]. We calculate the trend of a profession or SOC major group as the Spearman's rank correlation coefficient [74] of its media frequency against time. A significant positive correlation denotes an increasing trend, and a significant negative correlation implies a decreasing trend over time ($\alpha = 0.05$).

Fig 6 shows the trend of the 23 major SOC groups, from most positive to most negative. Table 3 lists these groups and their frequency trends. Only 3 SOC groups showed an increasing frequency trend in mentions over time, while 16 SOC groups decreased in frequency over time. The rank correlation was not significant for the remaining 4 SOC groups. These SOC

**Table 3. SOC major groups with increasing, decreasing or no frequency trend over time.**

| Increasing Frequency | No Trend |
|---|---|
| Computer and Mathematical Occupations | Legal Occupations |
| Life, Physical, and Social Science Occupations | Food Preparation and Serving Related Occupations |
| Art, Design, Entertainment, Sports, Media Occupations | Healthcare Support Occupations |
| | Business and Financial Operations Occupations |
| **Decreasing Frequency** | |
| Personal Care and Service Occupations | Community and Social Service Occupations |
| Installation, Maintenance, and Repair Occupations | Healthcare Practitioners and Technical Occupations |
| Farming, Fishing, and Forestry Occupations | Protective Service Occupations |
| Architecture and Engineering Occupations | Production Occupations |
| Sales and Related Occupations | Transportation and Material Moving Occupations |
| Management Occupations | Office and Administrative Support Occupations |
| Educational Instruction and Library Occupations | Building, Grounds Cleaning and Maintenance Occupations |
| Construction and Extraction Occupations | Military Specific Occupations |

A SOC major group has increasing or decreasing frequency trend if its Spearman's rank correlation is significantly positive or negative respectively.

groups contain 500 professions (see section Taxonomy Creation). We analyze the trend of some professions belonging to these SOC groups.

**Computer and Mathematical Occupations**: This SOC group includes mathematicians and computer-related professions. Fig 7a) shows the frequency trend of two of its professions: hacker and programmer. Both occupations showed a positive frequency trend, but mentions of hackers increased more than programmers.

**Life, Physical, and Social Science Occupations**: This SOC group includes archaeologists, astronauts, biologists, chemists, geologists, etc. Almost all its professions showed an increasing frequency trend over time. Fig 7b) shows the trend of four occupations: geologist, biologist, anthropologist, and economist. Mentions of geologists and biologists are consistently more frequent than economists and anthropologists.

**Arts, Design, Entertainment, Sports, and Media Occupations**: This SOC group includes entertainment, sports, music, and media-related professions. Fig 7c), 7d), 7e) and 7f) show the frequency trends of some of its professions. Mentions of actor dominate actress mentions in media content. The word actor can be used as a gender-neutral term, explaining part of this trend. The frequency of sports-related professions dipped in the 1960s but has increased since then. Pianist mentions decreased, whereas mentions of bass players, guitarists, and drummers increased over time. Mentions of journalists and reporters are more frequent than correspondents and columnists.

**Legal Occupations**: Legal occupations include lawyers, judges, attorneys, prosecutors, etc. Fig 7g) shows the frequency trend of defense attorneys and prosecutors. Mentions of prosecutors (a lawyer who conducts a case against a defendant) are more frequent than defense attorneys (a lawyer who defends the client against criminal charges).

**Food Preparation and Serving Related Occupations**: This SOC group includes professions related to food serving and food preparation. Fig 7h), 7i) and 7j) show the frequency trends of some of its occupations. Mentions of waiters are more frequent than waitresses overall. Similar to actors, waiters can refer to either gender, so this trend is not surprising. However, mentions of waiters have decreased over time, whereas mentions of waitresses have

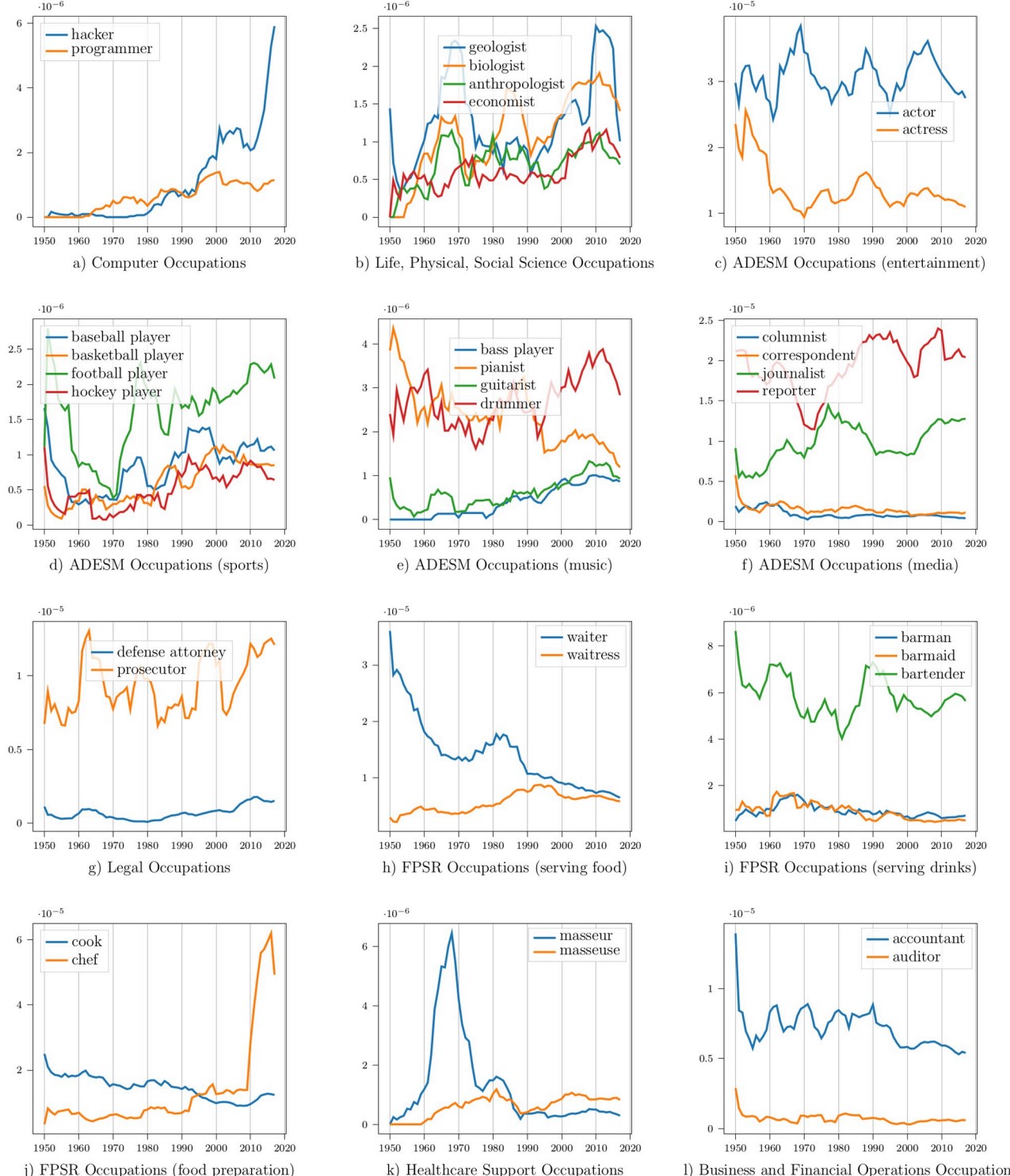

**Fig 7. Frequency trends of different professions over time in media subtitles.** ADESM = Arts, Design, Entertainment, Sports, and Media.
FPSR = Food Preparation and Serving Related.

increased. Gendered professional mentions like barman and barmaid are less frequent than the gender-neutral term: bartender. Mentions of cooks were more common than chefs, but the latter became more frequent in media content in the 2010s.

**Healthcare Support Occupations**: This SOC group includes healthcare assistants, nursing aides, massage therapists, etc. Fig 7k) shows the frequency trend of two gendered professions: masseur (male) and masseuse (female). The frequency of masseurs has significantly decreased over time after it peaked around 1970. Mentions of masseuses have become more common than masseurs. The frequency of the gender-neutral term, massage therapist, has increased over time.

**Business and Financial Operations Occupations**: This SOC group includes accountants, contractors, auditors, etc. Fig 7l) shows the frequency trend of accountants and auditors. Their frequencies have mostly remained steady over time. Accountants are more frequent than auditors.

**Personal Care and Service Occupations**: This SOC group includes barbers, valets, nannies, ushers, etc. Fig 8a) shows the frequency trend of three child-care-related professions: nanny, babysitter, and governess. The mention frequency of nannies and babysitters has increased over time. The word governess is an archaic term, and its usage has declined.

**Installation, Maintenance, and Repair Occupations**: This SOC group includes mechanics, electricians, locksmiths, etc. Fig 8b) shows the frequency trend of electricians and mechanics. The mentions of both these professions have decreased in media subtitles.

**Farming, Fishing, and Forestry Occupations**: This SOC group includes farmers, shepherds, herders, fishermen, etc. Fig 8c) shows the frequency trend of farmers, fishermen, and hunters. Mentions of farmers and fishermen have decreased, whereas mentions of hunters have increased over time.

**Architecture and Engineering Occupations**: This SOC group includes architects, designers, engineers, surveyors, etc. Fig 8d) shows the frequency trend of engineers and architects. The frequency of architect mentions has remained steady over time, whereas mentions of engineers have diminished.

**Sales and Related Occupations**: This SOC group includes sales and real-estate-related professions. Fig 8e) and 8f) show the frequency trends of some of its occupations. Mentions of retailers, distributors, vendors, and brokers have increased over time. The increase in frequency is even more prevalent in real-estate jobs: estate agent, realtor, and real estate agent.

**Management Occupations**: This SOC group includes administrative professions and occupations related to the management of educational institutions. Fig 9g) and 9h) show the frequency trend of some management occupations. Mentions of both congressman and congresswoman have increased over time. Although congressmen are mentioned more than congresswomen, the frequency of congresswomen has increased at a higher rate (not discernible from the graph). Mentions of senators, governors, and mayors have decreased over time. The frequency of headmistress mentions has increased, whereas headmaster mentions have decreased in media content. The frequency of the gender-neutral term, principal, has increased.

**Educational Instruction and Library Occupations**: This SOC group includes teachers, professors, librarians, etc. Fig 8i) shows their frequency trends. Professor mentions have decreased greatly over time, whereas the frequency of teacher and librarian mentions have increased slightly.

**Construction and Extraction Occupations**: This SOC group includes builders, carpenters, masons, miners, etc. Fig 8j) shows their frequency trends. The frequency of masons has remained steady, but miner and carpenter mentions have reduced over time.

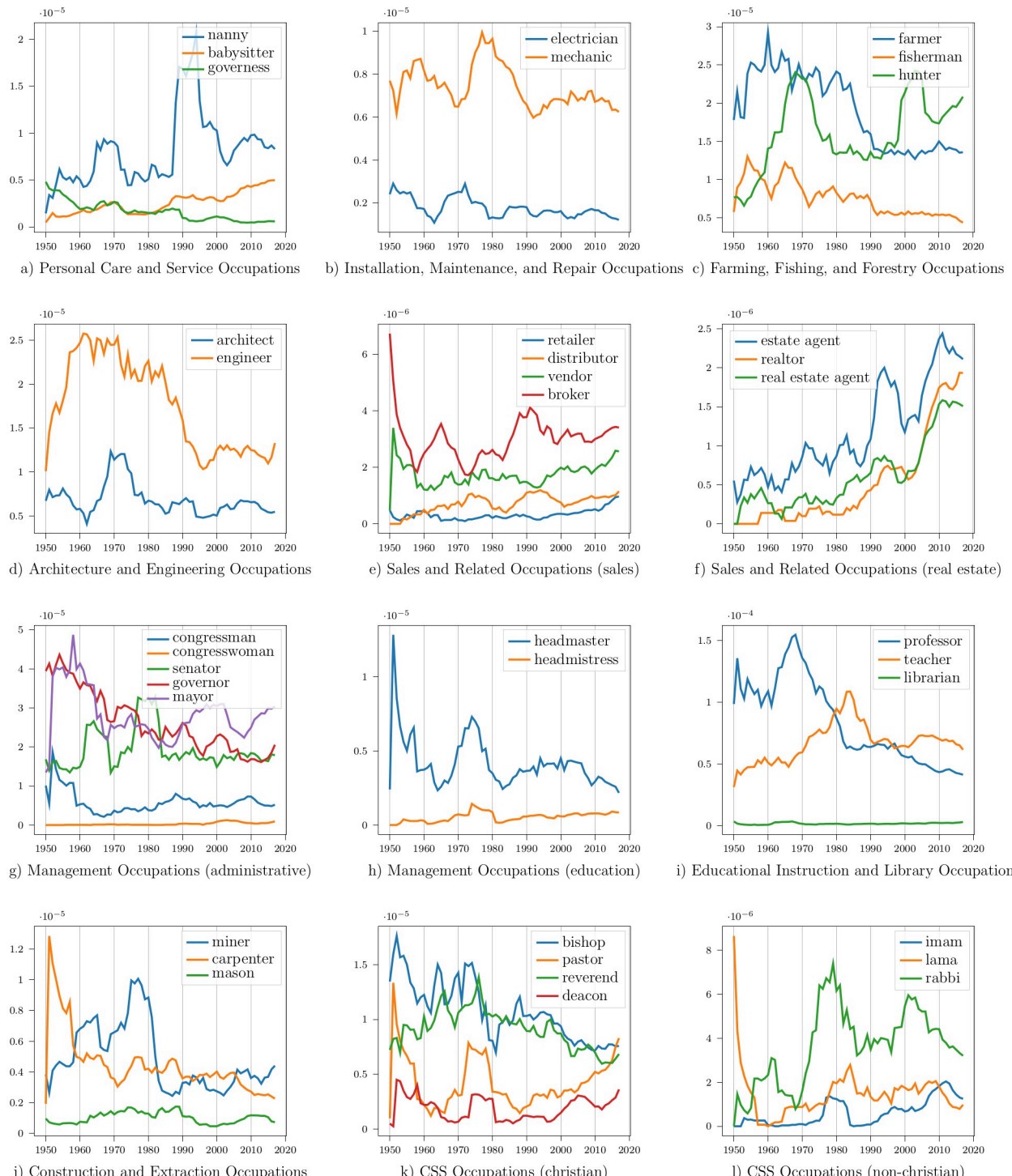

**Fig 8. Frequency trends of different professions over time in media subtitles.** CSS = Community and Social Service.

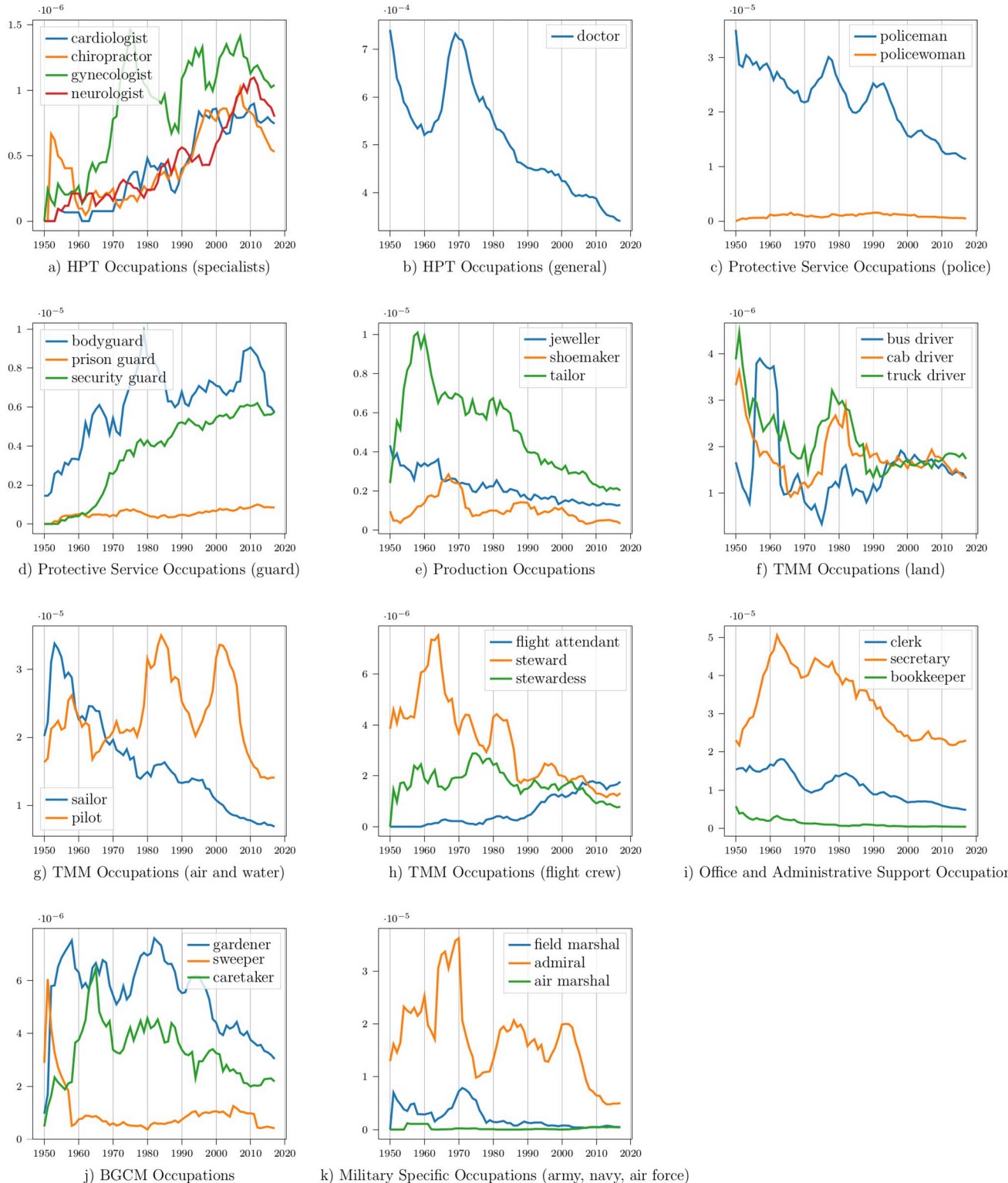

**Fig 9. Frequency trends of different professions over time in media subtitles.** HPT = Healthcare Practitioners and Technical. TMM = Transportation and Material Moving. BGCM = Building and Grounds Cleaning and Maintenance.

**Community and Social Service Occupations**: This SOC group includes religious and social workers. Fig 8k) and 8l) shows the frequency trend of some religious occupations. The mention frequency of bishops and reverends has decreased over time, whereas the frequency of pastors and deacons has increased. Mentions of imams, lamas, and rabbis have also increased over time, but the frequency of priests has decreased.

**Healthcare Practitioners and Technical Occupations**: This SOC group includes healthcare-related professions. Fig 9a) and 9b) show the frequency trend of some of its professions. Mentions of healthcare specialists like cardiologists, chiropractors, gynecologists, and neurologists have increased over time, but the frequency of the generic term, doctor, has decreased.

**Protective Service Occupations**: This SOC group includes law enforcement and protective service workers. Fig 9c) and 9d) show the frequency trend of some of its professions. Mentions of policemen have decreased over time but are still much higher than mentions of policewomen, which have remained steady. Guard occupations like bodyguards, prison guards, and security guards have become more frequent in media content.

**Production Occupations**: Production occupations include artisans, cobblers, jewelers, millers, etc. Fig 9e) shows the frequency trend of some of its professions: jeweler, shoemaker, and tailor. The mention frequency of all three professions has decreased over time.

**Transportation and Material Moving Occupations**: This SOC group includes professions related to the transportation of goods and people. Fig 9f), 9g) and 9h) show frequency trends of different transportation-related professions. Mentions of bus drivers increased, cab drivers remained steady, and truck drivers decreased over time. Mentions of sailors decreased more rapidly than pilots. The frequency of gendered professional terms like steward and stewardess decreased over time. The frequency of the gender-neutral term, flight attendant, increased.

**Office and Administrative Support Occupations**: This SOC group includes clerks, receptionists, tellers, notaries, etc. Fig 9i) shows the frequency trend of some of its professions: clerk, secretary, and bookkeeper. The frequency of all three professions decreased over time.

**Building, Grounds Cleaning and Maintenance Occupations**: This SOC group includes construction workers, maids, chamberlains, janitors, etc. Fig 9j) shows the frequency trend of some of its professions: gardener, sweeper, and caretaker. Mentions of all three professions decreased over time in media content.

**Military Specific occupations**: Military occupations include army, naval and air-force-related professions. Fig 9k) shows the frequency trend of some of its professions. Mentions of field marshal (army) and admiral (navy) decreased over time. The frequency of air marshal mentions increased (not discernible from the graph).

The frequency trend of the SOC group does not reflect the frequency trend of all its professions. For example, Fig 8e) and 8f) show increasing trends for many sales-related professions, but the overall frequency of the Sales Occupations SOC group decreased. A large proportion of Sales Occupations mentions are comprised of bankers and cashiers, whose frequencies have decreased.

## Profession sentiment

We find the sentiment expressed toward each professional mention in our dataset using the LCF model (see section Profession Search). The computed sentiment can take the following values: positive, negative, or neutral. We call all mentions tagged with non-neutral sentiment as opinionated mentions. We represent the sentiment expressed towards a profession or a major SOC group as the number of positive sentiment mentions divided by the total number of opinionated mentions. We find the sentiment trend of professions by calculating the Spearman's rank correlation [74] between the proportion of positive sentiment mentions and time.

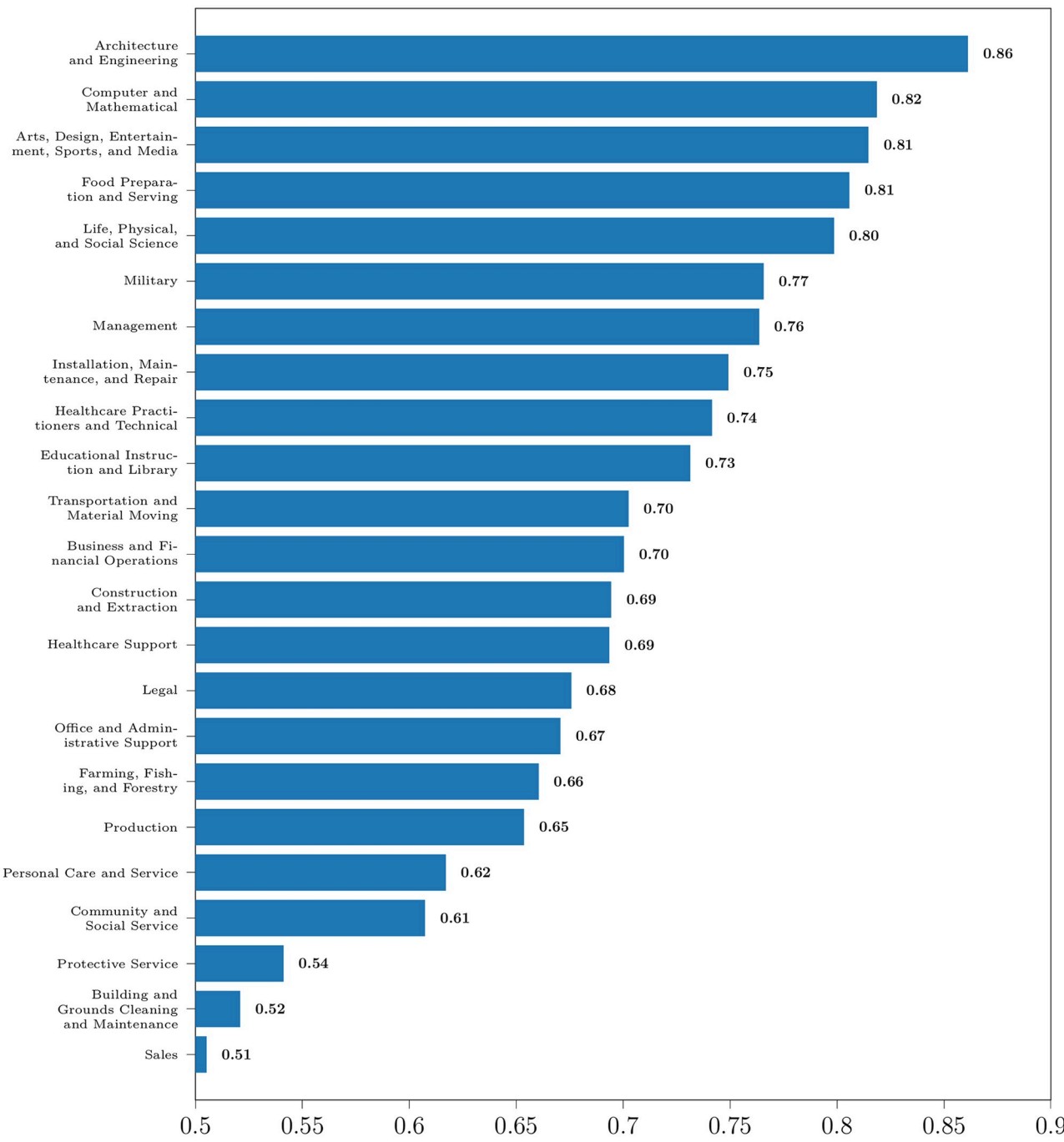

**Fig 10. Proportion of positive sentiment mentions in opinionated mentions of the SOC groups.**

Fig 10 shows the proportion of positive sentiment mentions of the 23 major SOC groups. From the figure, we observe that the proportion is always greater than 0.5. Therefore, the number of positive sentiment mentions is greater than the number of negative sentiment mentions for all the SOC groups. This might be because of an inherent positive bias in the sentiment of media narratives. Architects and engineers are talked about most positively, and sales-related

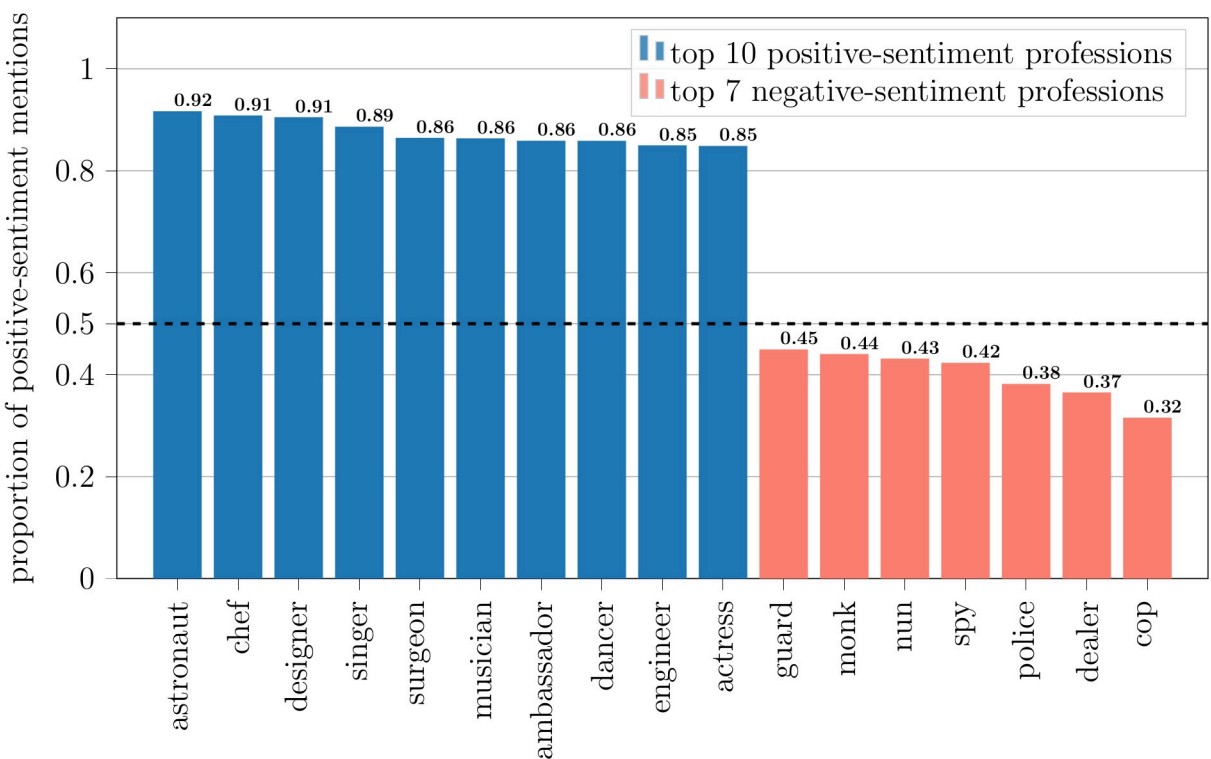

**Fig 11. Top 10 positive-sentiment and top 7 negative-sentiment professions.** The blue-colored professions have the top 10 highest proportion of positive sentiment mentions. The red-colored professions have the top 10 highest proportion of negative sentiment mentions.

occupations are talked about most negatively. STEM professions are generally expressed in positive sentiment. Four out of the top ten SOC groups ordered by the proportion of positive sentiment contain STEM professions: Architecture and Engineering; Computer and Mathematical; Life, Physical, and Social Science; and Healthcare Practitioners and Technical Occupations. Professions involving manual labor are generally expressed with negative sentiment. Four out of the bottom ten SOC groups contain blue-collar jobs: Building, Grounds Cleaning and Maintenance; Production; Farming, Fishing, and Forestry; and Construction and Extraction Occupations.

We analyze the proportion of positive sentiment mentions for individual professions. Fig 11 shows the top ten professions with the highest proportion of positive sentiment mentions and the bottom seven professions with the lowest proportion of positive sentiment mentions from the list of most occurring top 100 professions in our subtitle corpus. More than half of the opinionated mentions of police and cops are negative. Religious workers like monks and nuns are talked about more negatively than positively. More than four-fifths of the mentions of singers, musicians, and dancers are positive. STEM professions like astronauts and engineers are also mentioned positively in media content.

We also study the sentiment trend of professions in the analyzed media content. Fig 12 shows the Spearman's rank correlation coefficient of the proportion of positive-sentiment mentions over time for some of the professions. The figure lists the top six professions with the highest correlation and the bottom ten professions with the lowest correlation from the list of most occurring top 100 professions in our subtitle corpus. The sentiment expressed towards therapists, spies, rangers, and detectives in media content trends toward becoming more

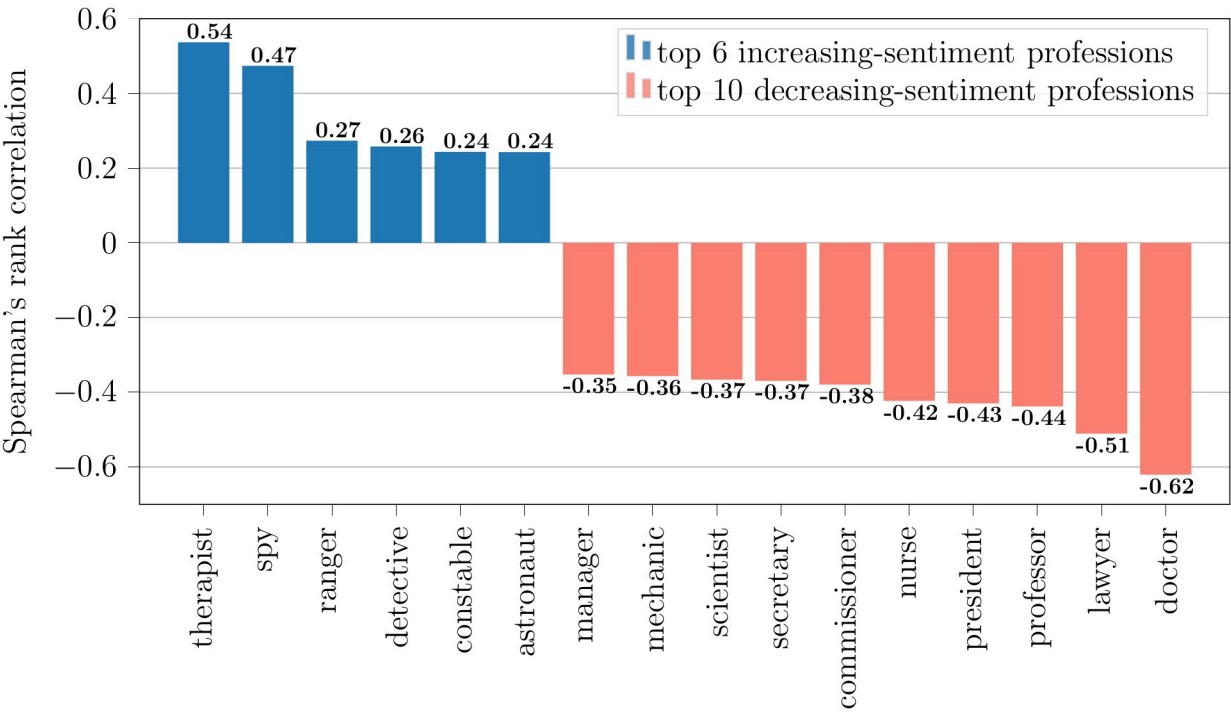

**Fig 12. Professions with the top increasing and top decreasing sentiment trends.** The blue-colored professions have the top 6 most positive Spearman's rank correlation of proportion of positive mentions vs time. Therefore, the proportion of positive sentiment mentions shows increasing trend over time for these professions. The red-colored professions have the top 10 most negative correlation and their proportion of negative sentiment mentions is increasing over time.

positive over time, whereas the sentiment expressed toward doctors, lawyers, professors, and scientists trends more negative. Lawyer mentions have the second-lowest sentiment correlation over time, behind doctors. Astronauts not only have the highest proportion of positive sentiment mentions overall (see Fig 11), but also one of the most positive sentiment trends.

## Media attributes

We have observed that both the frequency of professional mentions and the sentiment expressed towards them change over time. In this section, we study the relationship of the following media attributes: year, genre, title type, and country of production (see section OpenSubtitles) with the observed media frequency and sentiment trends of professions and SOC major groups.

We perform a regression analysis to analyze the effect of media attributes. The response is the frequency or sentiment of the profession or the SOC major group. The predictors are year, genre, title type, and country of production. All predictors are categorical variables except for year, which is numeric. Genre and country are multi-valued. An IMDb title can belong to multiple genres, and its production can take place in multiple countries. Therefore, we create a categorical variable for each possible genre and country, taking values 0 or 1. We set it to 1 if the IMDb title belongs to the corresponding genre or its production takes place in the corresponding country. We ignore media attribute configurations for which the number of IMDb titles is less than 30. We use logistic regression because both frequency and sentiment are proportions, bounded between 0 and 1 (defined in sections Profession Frequency and Profession

Sentiment). We use a generalized linear model with the binomial family and logit link. We provide the total number of ngrams and opinionated mentions (defined in section Profession Sentiment) as prior weights to the model to specify the number of trials when the response is frequency or sentiment, respectively.

Figs 13 and 14 show the significant predictors ($\alpha$ = 0.05) when the response is the frequency of professions and SOC groups, respectively, in the form of a heatmap. The color of the cells indicates the sign of the coefficient (blue = +, red = −) and the intensity of the color denotes its magnitude. The white cells mean that the predictor is not significant. We observe some interesting relationships between the frequency of professional mentions and media attributes. The frequency of actors increases, but the frequency of actresses decreases when the genre is adventure, documentary, or thriller. The reverse is true when the genre is romance. Mentions of lawyers increase in crime, drama, and mystery genre media content. The frequency of lawyers and attorneys increase, and the frequency of prosecutors decreases when the country of production is the United States. United States-produced movies and TV shows mention cops and sheriffs more than inspectors and police. The opposite is true for United Kingdom-produced titles. The frequency of detectives and spies increases in mystery genre titles. Comedy, reality-TV, and music genres increase the frequency of dancers, singers, and artists. The frequency of doctors, nurses, and surgeons in movies is higher than in TV shows. In science-fiction and family media titles, the frequency of doctors increases, and the frequency of nurses and surgeons decreases. Documentary titles increase the frequency of reporters and journalists. Mentions of engineers and scientists increase when the genre is science-fiction or documentary and decreases in comedy and fantasy genres. The frequency of teachers and professors decreases in action and adventure genres. Movies mention teachers more than TV shows, but the opposite holds for professors. Mentions of senators, mayors, and presidents increases in news and thriller genres. The frequency of lieutenants and soldiers increases in action and war media titles and decreases when the genre is fantasy and romance.

Fig 14 shows the coefficient heatmap of media attributes when the response is the frequency of SOC groups. We highlight some relationships between SOC frequency and media attributes. The frequency of Management, Business, and Financial Operations occupations increases in biography and news genre media titles. Documentaries and science-fiction movies and TV shows frequently mention STEM professions like Computer, Mathematical, Architecture, Engineering, Life, Physical, and Social Science occupations. Mentions of Community and Social Service occupations decrease in reality TV shows, sports, and family movies. The frequency of Legal occupations increases in news genre titles. The frequency of Arts, Design, Entertainment, Sports and Media occupations increases in music, sports, game show, and biography genre titles. Mentions of Healthcare Practitioners increase in news and drama genres and decrease in musicals. Food Preparation and Serving related professions occur highly in reality TV shows and decrease in music and adventure movies. The frequency of manual labor jobs like Construction, Extraction, Production, Building, Grounds Cleaning, and Maintenance occupations decreases when the country of production is the United States. Mentions of Military occupations increase in war and action movies and decrease in comedy and family shows.

We study the effect of media attributes on the proportion of positive sentiment mentions of professions and SOC Groups. Unlike frequency, the number of significant predictors are fewer. Therefore, instead of a heatmap, we tabulate the significant media attributes with the most positive and negative coefficients ($\alpha$ = 0.05) in Table 4. The proportion of positive sentiments expressed towards marshalls, mayors, professors, and prosecutors increases when the genre is mystery. The proportion of negative sentiment increases for congressmen, priests, and prosecutors in crime genre media titles. The average sentiment of SOC groups containing STEM occupations like Computer, Mathematical, Life, Physical, and Social Science

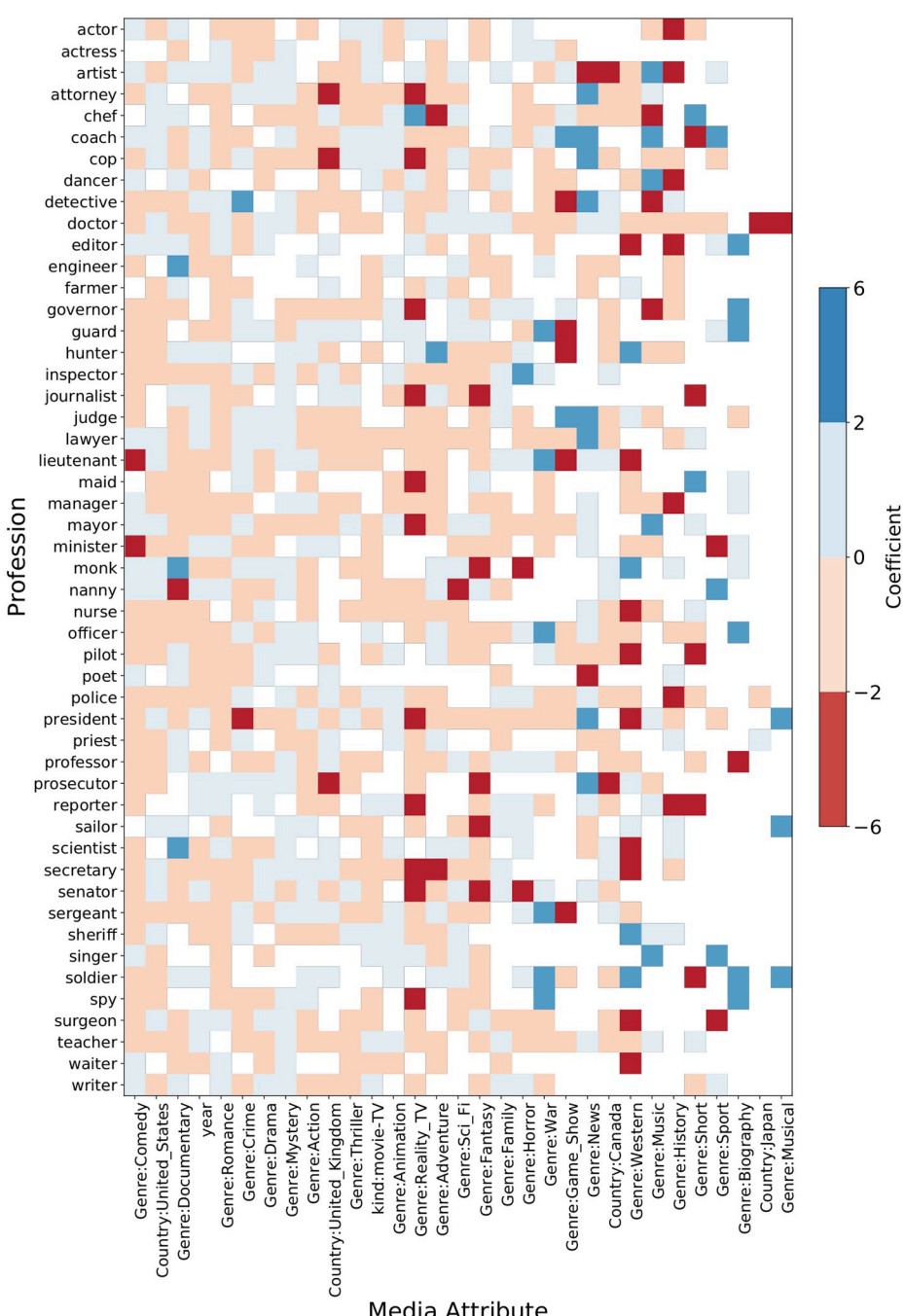

**Fig 13. Heatmap of coefficient values of media attributes in predicting profession frequency.** The color denotes the sign of the coefficient (blue = +, red = −) and the intensity of the color is proportional to the magnitude of the coefficient. The blank cells indicate that the media attribute is not a significant predictor for the frequency of the corresponding profession.

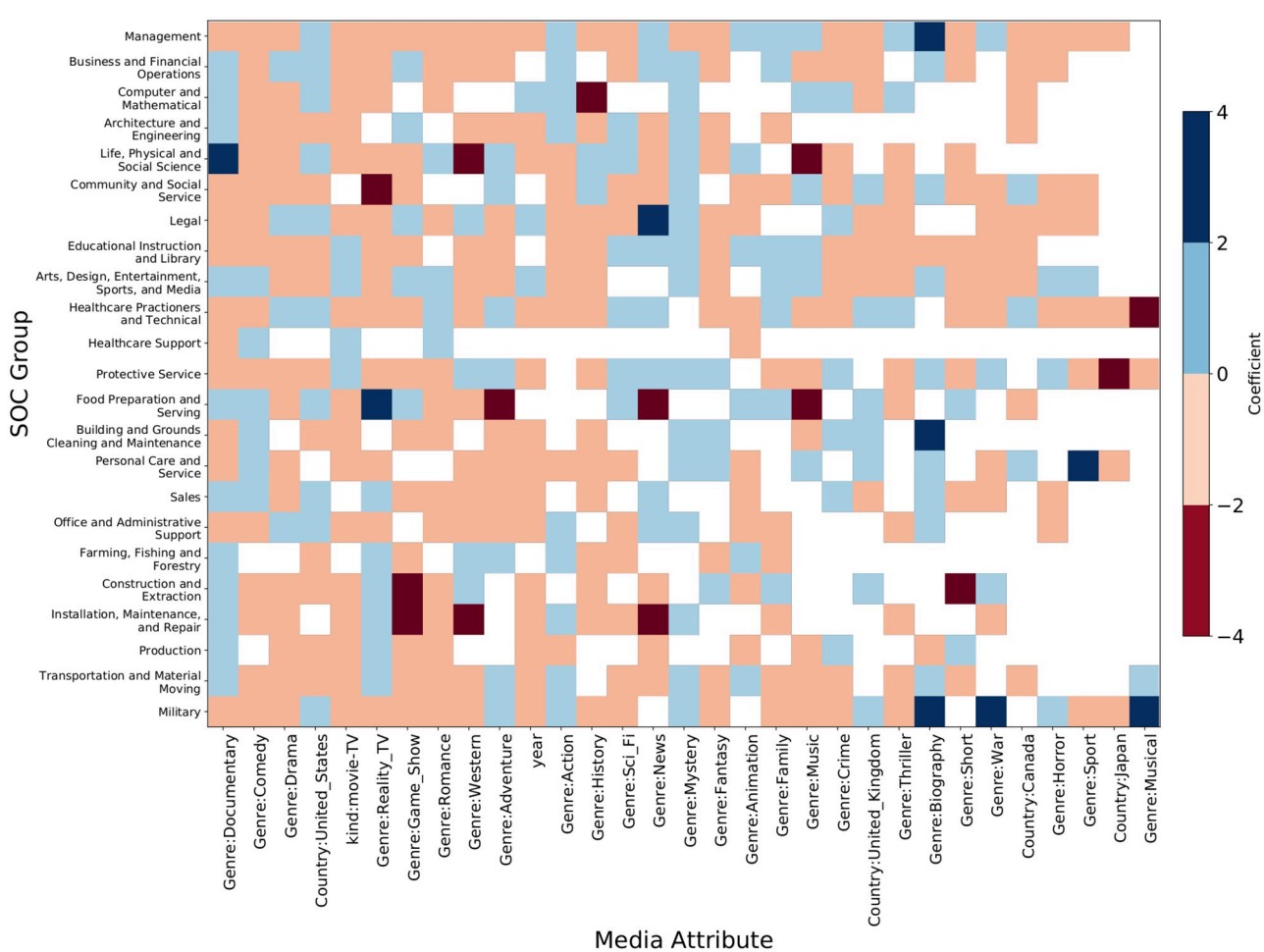

**Fig 14. Heatmap of coefficient values of media attributes in predicting SOC group frequency.** The color denotes the sign of the coefficient (blue = +, red = −) and the intensity of the color is proportional to the magnitude of the coefficient. The blank cells indicate that the media attribute is not a significant predictor for the frequency of the corresponding SOC group.

occupations becomes more positive in documentaries. Movies express more negative sentiment than TV shows towards Legal occupations.

## Employment

We analyzed the frequency and sentiment trends of different professions and SOC groups in media content. We also studied the effect of media attributes on these representations. However, our analysis has been limited largely to the entertainment media domain, by using NLP on the subtitles of media content. In this section, we study the correlation between media frequency and real-world events, namely employment figures. We obtained the employment data of SOC major groups from the Occupational Employment Statistics survey (OES). The survey does not provide employment numbers for individual professions. Therefore, we only conduct our analysis on SOC groups. We calculate the Spearman's rank correlation [74] between the media frequency of the SOC group and the proportion of the working population employed in any of the professions of the SOC group. We compute the correlation for the period 1999-2017. The employment data for the earlier years is not available.

**Table 4. Effect of media attributes on the sentiment of professions and SOC groups.**

| Profession / SOC Group | Media Attributes | |
|---|---|---|
| | **Positive Coefficients** | **Negative Coefficients** |
| actor | | Genre:Drama |
| actress | | Country:United Kingdom |
| architect | Genre:Documentary | |
| artist | Genre:Game Show | Genre:Fantasy |
| banker | | kind:movie-TV |
| chef | Genre:Game Show | Country:United Kingdom |
| congressman | | Genre:Crime |
| cop | Genre:Adventure | Genre:Family |
| dealer | Country:United States | kind:movie-TV |
| detective | Genre:Crime | |
| district attorney | | kind:movie-TV |
| doctor | Genre:Action | Genre:Horror |
| engineer | Genre:Documentary | kind:movie-TV |
| judge | | kind:movie-TV |
| lawyer | Genre:Romance | Genre:Thriller |
| manager | Genre:Family | |
| marshall | Genre:Mystery | |
| mayor | Genre:Mystery | |
| musician | | Genre:Comedy |
| nurse | | Genre:Documentary |
| officer | Country:United States | Genre:Fantasy |
| police | Genre:Animation | Genre:News |
| priest | Genre:Thriller | Genre:Crime |
| professor | Genre:Mystery | Genre:Adventure |
| prosecutor | Genre:Mystery | Genre:Crime |
| scientist | | Genre:Reality TV |
| secretary | Genre:Action | |
| sheriff | | Genre:Action |
| social worker | Country:United Kingdom | |
| soldier | Genre:Western | |
| spy | Genre:Animation | Country:United Kingdom |
| Architecture and Engineering | Genre:Documentary | kind:movie-TV |
| Arts, Design, Entertainment, Sports, and Media | Genre:Game Show | Genre:Crime |
| Building and Grounds Cleaning and Maintenance | Genre:Documentary | Genre:Crime |
| Business and Financial Operations | Genre:Adventure | Genre:News |
| Community and Social Service | Genre:Mystery | Genre:Family |
| Computer and Mathematical | Genre:Documentary | |
| Construction and Extraction | | Genre:Sci Fi |
| Educational Instruction and Library | Genre:Documentary | Genre:Sport |
| Farming, Fishing and Forestry | | Genre:Western |
| Food Preparation and Serving | Genre:Documentary | Genre:Crime |
| Healthcare Practitioners and Technical | Genre:Action | Genre:Horror |
| Legal | Genre:Romance | kind:movie-TV |
| Life, Physical and Social Science | Genre:Documentary | Genre:Reality TV |
| Management | Genre:War | Genre:Crime |
| Military | Genre:Western | Genre:Comedy |

*(Continued)*

**Table 4.** (Continued)

| Profession / SOC Group | Media Attributes | |
|---|---|---|
| | **Positive Coefficients** | **Negative Coefficients** |
| Personal Care and Service | Genre:Documentary | Genre:Animation |
| Production | Genre:Action | Genre:Fantasy |
| Protective Service | Genre:Reality TV | Genre:News |
| Sales | Genre:Horror | Genre:Crime |
| Transportation and Material Moving | | Genre:History |

This table lists the media attributes with the highest significant positive and negative coefficients in predicting the sentiment of professions and SOC major groups.

Fig 15 shows the correlation between media frequency and employment for the SOC groups, from most positive to most negative. The correlation is positive for 14 out of the 22 SOC groups (64%) and negative for the rest. Therefore, the trend of media frequency correlates with the employment trend for most SOC groups. The number of media mentions of the SOC group increases as more people are employed in its professions, and decreases as people move away to other occupations. The correlation is significantly negative for only two SOC groups: Healthcare Practitioners and Social Service occupations.

## Discussion

The frequency and sentiment expressed towards professionals vary in media content depending on the character archetype, the story being told, year of production, and genre, among other variables. We developed a visualization tool [75] to examine the frequency, sentiment, media attribute, and employment trends in our subtitle corpus. While examining the wealth of results using this tool, a few general findings and specific trends stood out to us. In this section, we highlight these trends, discuss some reasons that may be suggestive for explaining the observed results, and try to reconcile these findings with those in relevant studies, both from the real world and the media domain. We underscore that our interpretation is purely speculative and requires careful, controlled experiments and surveys to further validate the claims. Also, this discussion is by no means an effort to explain all our findings, and we hope future research in this space may further explain some of the results we found in this large-scale analysis.

### Findings from frequency analysis

We observed that gender-neutral terms like massage therapists and flight-attendants are becoming more frequent than their gendered counterparts. We suggest that this trend is due to the increased awareness stemming from conversations about gender-neutral terms among the youth, parents, and the LGBTQ+ community [76, 77]. Analyzing explicit gendered profession terms, we observed that the frequency of some female job titles such as waitresses, congresswomen, and policewomen has either increased or remained steady relative to the corresponding male job titles (i.e., waiters, congressmen, and policemen). Equal opportunity legislation, women's rights movements [78], and constitutional amendments for women's suffrage [79] have improved the representation of women in police and government. This may have driven a similar trend of increasing female representation in media content. While there is a change in trend between the explicitly gendered job titles over the years, the overall frequency of most male job titles exceeds their female counterparts.

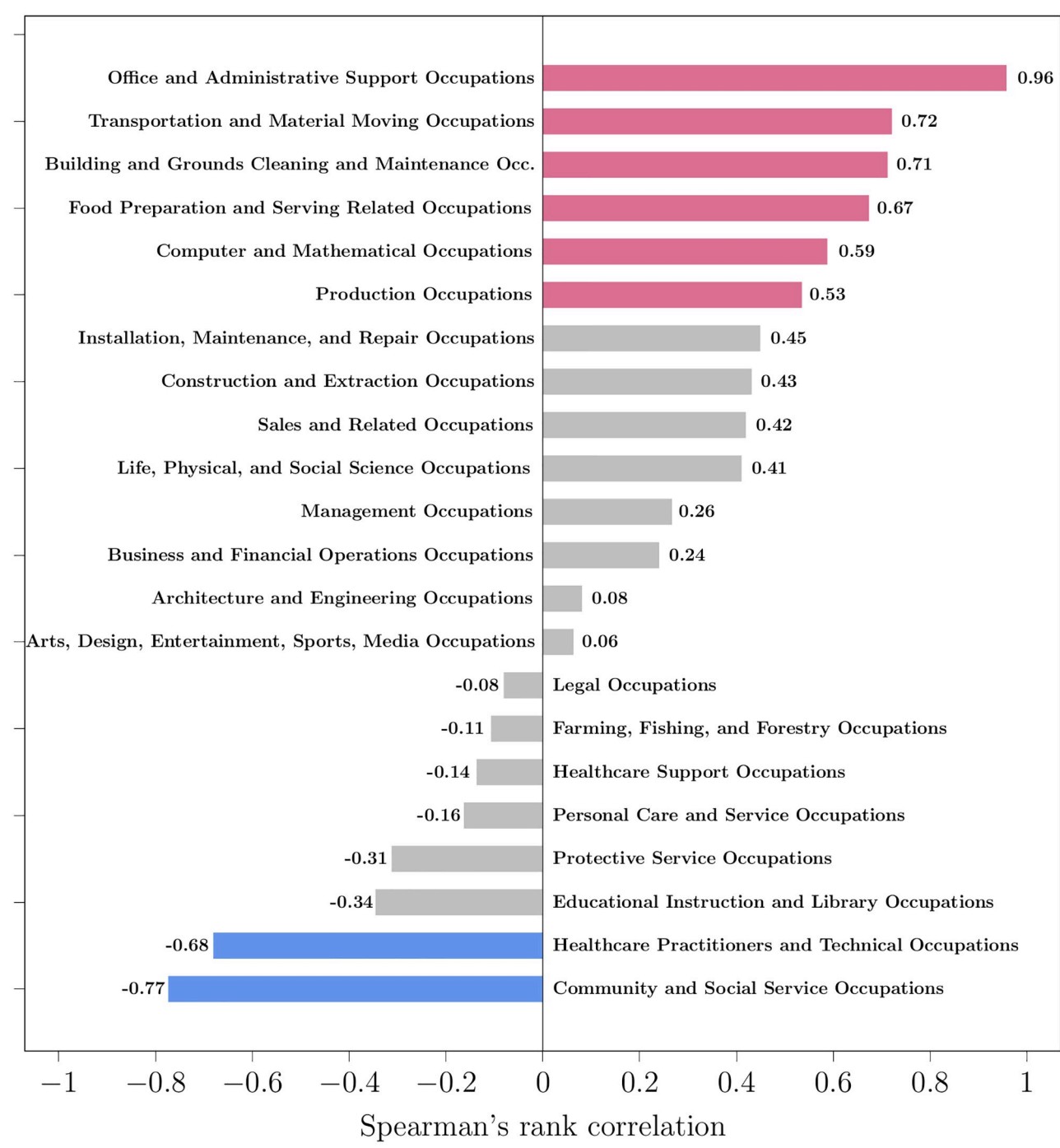

**Fig 15. Spearman's rank correlation coefficient of the media frequency of SOC groups with employment.** The red-colored bars have positive correlation: media frequency and employment have the same trend. The blue-colored bars have negative correlation: media frequency and employment have opposite trends. The grey-colored bars mean that the correlation is not statistically significant ($\alpha = 0.05$).

The frequency of STEM, sports, arts, and design occupations has increased, whereas the frequency of construction, farming, and manual labor jobs has decreased. Improvements in cinematography and visual effects technology, space travel, the emergence of the world wide web, and sports documentaries have contributed to the increased fascination of science and sports

in media [80, 81]. These advances and interests potentially explain the increase in the frequency of STEM-related and sports portrayals on the big screen. The frequency of specialized professions like cardiologists, gynecologists, and neurologists has increased, but the frequency of generic terms like doctors and nurses has decreased. The evolution of medicine and research, students' preference for higher-paying specialties [82], and the emergence of novel diseases have increased the demand for specialized professions over general ones in the medical field [83]. We posit that as new and specialized professions become popular career choices and essential in catering to the needs of the continuously evolving society, they also attract the interest of media creators.

Thus far, we have discussed the frequency trends of individual professions. It is important to note that this does not always reflect the overall trend of the subsuming SOC group. Sales-related occupations like retailers, vendors, brokers, and realtors show an increase in frequency, but the aggregate frequency of the associated SOC group shows a decreasing trend. Technology improvements in computerized financial transactions (barcode scanning, automatic teller machines, and mobile banking) and customers' preference for self-service checkouts probably contributed to the decrease in employment of bankers and cashiers [84, 85], or at least their explicit mentions in media narratives.

### Findings from sentiment analysis

STEM occupations are favorably (i.e., positive sentiment) mentioned in the subtitles corpus we analyzed. However, the sentiment expressed toward professions that predominantly involve manual labor—the so-called "blue-collar" jobs—is largely negative. Overall, police, monks, and nuns are mentioned negatively, whereas musicians and engineers are mentioned positively.

The sentiment expressed toward therapists, astronauts, and detectives was observed to trend more positive over time, while the sentiment toward doctors, lawyers, professors, and presidents, was more negative. The decreasing trend of positive sentiment toward lawyers is consistent with Asimow's work [16] that showed the decline in public opinion of lawyers. This may be one of the explanatory factors behind the increasingly negative portrayal of lawyers in movies. In contrast, the negative sentiment expressed towards doctors, scientists, and police that we found in our media corpus does not appear to agree with the findings of public polls. Several studies over the years [86–88] show that people usually trust these professions. We note that public opinion also depends on several social and demographic factors such as political leanings and the level of educational attainment [87, 88]. Factors like the popular stereotype of the "mad scientist" (someone who is more concerned about their scientific findings than human welfare) [18] in movies, grievances of the health care system [89], and general distrust towards people and institutions in positions of authority, might have contributed towards these negative portrayals in media.

We have made several hypotheses to explain the observed frequency and sentiment trends and require controlled experiments to verify these claims. It is challenging to design such studies because we are trying to relate media narratives to real-world events—two very different domains. As of now, we are unsure regarding what types of experiments are required, but we suggest some ideas. We can survey scriptwriters and movie directors to find what societal events inspired their stories and creative decisions. We can also try to map change points in media trends to real-world events to explain their possible cascading effects on media narratives and genres. We leave this task for future research.

### Findings from media attribute analysis

Media attributes like genre, country of production, and content type affect the frequency and sentiment trends of professions in our subtitle corpus. Genre, in particular, seems to be a good predictor of the type of profession mentioned in the subtitles. For example, science-fiction movies often mention engineers and scientists, action and war genres mention lieutenants and soldiers, and mystery titles contain detectives and spies. Adventure/thrillers contain more references to male actors than female actors. However, the opposite is true for romantic movies. Consistent with findings in previous work [90], this suggests that gender bias is prevalent in these media genres.

In examining the country of production, we found that sheriffs are mentioned more than inspectors in titles produced in the US, and the opposite holds for movies produced in the UK, reflecting the differences in English usage, and law enforcement structures in these countries. Regarding the type of content, we did not find significant differences in the mentions of job titles between movies and TV shows in our subtitle corpus.

### Findings from employment analysis

Interestingly frequency of job titles in media correlates with real-world employment statistics of corresponding professions. We observed a significant positive correlation between media frequency and employment trends of most SOC groups. Professions that typically employ more people were also more frequently mentioned in media content. We note that these correlation findings do not imply causality; a question that requires further systematic study.

### Limitations

There are several limitations of the present work. First, subtitles provide only a limited view of the complete movie or TV show. Features, such as the character's behavior on screen, their interactions with other people, and their character arc can only be studied by carefully viewing the entire movie or TV episode. Therefore, our representation measures of frequency and sentiment of job titles in subtitles do not capture all aspects of their media portrayal. Second, our frequency and sentiment analyses do not control for the frequency or sentiment expressed toward these professions in everyday language. For example, the frequency of policewomen might have been less than the frequency of policemen in media subtitles because that is how their usage evolved in literature. A possible solution is to use the corresponding Google ngram frequencies [23] of the professional words as covariates in our analysis. Third, our study has primarily focused on the representation of professions in the US and the UK (see the distribution of production country in section OpenSubtitles). One needs to adapt our proposed taxonomy and models for different cultures and languages. Our profession taxonomy does not contain all job titles in other countries. Depending on the application of this taxonomy, one may have to add professions specific to a local region for a comprehensive study.

### Conclusion

In this work, we have created a searchable taxonomy of professions to facilitate job title search in short context documents like media subtitles. We used WordNet synsets and word sense disambiguation methods to retrieve professional mentions in movie and TV show subtitles. We automatically classified the sentiment (positive, negative, or neutral) expressed towards these professional mentions in the subtitle sentence. We analyzed the frequency and sentiment trends of professions and SOC groups, the effect of media attributes on these trends, and showed that media frequency of professions correlates with their employment statistics. Future

work entails extending our analysis to include industries and businesses, and to explore other media domains like news and social media. Importantly, future work should also consider investigating causal relations, beyond correlations, between media representations and employment trends.

The profession taxonomy and sentiment-annotated subtitle corpus is publicly available [26]. We have also released a visualization tool to explore and view the trends in our subtitle corpus [75]. We hope to add more features and better infographics to the tool in the future.

## Author Contributions

**Conceptualization:** Sabyasachee Baruah, Krishna Somandepalli.

**Data curation:** Sabyasachee Baruah.

**Formal analysis:** Sabyasachee Baruah.

**Funding acquisition:** Shrikanth Narayanan.

**Investigation:** Sabyasachee Baruah.

**Methodology:** Sabyasachee Baruah, Krishna Somandepalli.

**Project administration:** Sabyasachee Baruah.

**Software:** Sabyasachee Baruah.

**Supervision:** Shrikanth Narayanan.

**Validation:** Sabyasachee Baruah, Krishna Somandepalli.

**Visualization:** Sabyasachee Baruah.

**Writing – original draft:** Sabyasachee Baruah.

**Writing – review & editing:** Sabyasachee Baruah, Krishna Somandepalli, Shrikanth Narayanan.

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
