## [Decision Letter · Decision Letter 0]

6 Jan 2022

PONE-D-21-27904Representation of professions in entertainment media: Insights into frequency and sentiment trends through computational text analysisPLOS ONE

Dear Dr. Baruah

Thank you for submitting your manuscript to PLOS ONE, and apologies once again for the length of time it took to send this work out for review. I am pleased to report that I have now managed to secure two expert reviews of your work. Both reviewers feel that it the work has merit but does not fully meet PLOS ONE’s publication criteria as it currently stands. Therefore, we invite you to submit a revised version of the manuscript that addresses the points raised during the review process. Both reviewers agree that this is a worthwhile and carefully conducted study, something that I also sensed when reading the manuscript. All of us appreciate the clear writing style. Reviewer 1 raises some "high-level" issues, mostly around the interpretation of your findings. They are particularly concerned about possible causal relationships between the correlates that you identify in your work, and I agree that this needs some finessing in a revision. As Reviewer 1 states, if one wishes to offer a particular causal interpretation, evidence needs to be provided to support this; otherwise, one should instead emphasise the nature of the correlations and at most tentatively suggest causal hypotheses that could be further explored in future work. At the same time, Reviewer 1 notes that the manuscript presents a body of facts, but does not offer possible explanations for them - so clearly some speculation is to be appreciated, as long as it is presented as such. Reviewer 2 makes a similar observation and would appreciate a more extensive discussion of where your results fit within current theoretical frameworks and implications for future work. I also felt this would be necessary to include in a revision, and despite the brevity of Reviewer 2's report, it gets to the heart of the main issue standing in the way of an acceptance decision. My impression is that all comments of the reviewers are reasonable, not in conflict, and can be accommodated with relatively minor additions to the manuscript. Certainly it does not look as though more technical work is required, just a little more effort in discussing and interpreting the results so that the reader can better appreciate the implications of your findings (rather than the nature of the findings themselves, which is well explained). My preference wherever possible is to avoid a second round of review, which I think is feasible if you are able to address all the points raised by the reviewers primarily through changes to the manuscript and only occasionally with a rebuttal that makes a compelling case without recourse to lengthy technical detail. Please submit your revised manuscript by Feb 20 2022 11:59PM. If you will need more time than this to complete your revisions, please reply to this message or contact the journal office at plosone@plos.org. Please include the following items when submitting your revised manuscript:A rebuttal letter that responds to each point raised by the academic editor and reviewer(s). You should upload this letter as a separate file labeled 'Response to Reviewers'.A marked-up copy of your manuscript that highlights changes made to the original version. You should upload this as a separate file labeled 'Revised Manuscript with Track Changes'.An unmarked version of your revised paper without tracked changes. You should upload this as a separate file labeled 'Manuscript'.If applicable, we recommend that you deposit your laboratory protocols in protocols.io to enhance the reproducibility of your results. Protocols.io assigns your protocol its own identifier (DOI) so that it can be cited independently in the future. For instructions see: https://journals.plos.org/plosone/s/submission-guidelines#loc-laboratory-protocols. Additionally, PLOS ONE offers an option for publishing peer-reviewed Lab Protocol articles, which describe protocols hosted on protocols.io. Read more information on sharing protocols at https://plos.org/protocols?utm_medium=editorial-email&utm_source=authorletters&utm_campaign=protocols.

We look forward to receiving your revised manuscript.

Kind regards,

Richard A Blythe

Academic Editor

PLOS ONE

Journal Requirements:

Reviewers' comments:

Reviewer's Responses to Questions

**Comments to the Author**

1. Is the manuscript technically sound, and do the data support the conclusions?

Reviewer #1: Partly

Reviewer #2: Yes

2. Has the statistical analysis been performed appropriately and rigorously? 

Reviewer #1: Yes

Reviewer #2: I Don't Know

3. Have the authors made all data underlying the findings in their manuscript fully available?

Reviewer #1: Yes

Reviewer #2: Yes

4. Is the manuscript presented in an intelligible fashion and written in standard English?

Reviewer #1: Yes

Reviewer #2: Yes

5. Review Comments to the Author

Reviewer #1: PLOS ONE

PONE-D-21-27904

Representation of professions in entertainment media: Insights into frequency and sentiment trends through computational text analysis

General Comments

The paper presents a method for extracting information about trends in the frequency and sentiment words that describe various occupations in movies and TV shows. The frequency and sentiment information is extracted from the subtitles of 136,000 movies and TV shows, using Named Entity Recognition, WordNet, Word Sense Disambiguation, and Sentiment Analysis.

In general, the paper is clearly written and the methods used for analysis of the data are appropriate. The following review first discusses some high-level issues and then discusses some low-level details.

High-level Concepts

Abstract: "Professions that employ more people have increased media frequency, supporting our hypothesis that media acts as a mirror to society."

-- Let E be the levels of employment in various professions. Let F be the frequency of various words for professions. Let S be the sentiment of various words applied to professions. The sentence quoted here suggests that variation in E causes variation in F and S: that is, E is the real thing, whereas F and S are merely reflections (mirrors) of E. The paper makes it clear that E is correlated with F and S, but the paper does not demonstrate how the causal connection goes. There are (at least) three possibilities: (1) Variation in E causes variations in F and S. (2) Variations in F and S cause variations in E. (3) Variation in some other thing, X, causes variation in E, F, and S. All three of these causal connections seem possible to me, and it is not at all clear which is the actual connection. The authors need to either (A) be clear that they have only demonstrated correlation and not causation or (B) provide solid evidence that there is indeed a specific causal relation connecting E, F, and S. If the authors choose (A) (causation unknown), then they should remove all expressions, such as "acts as a mirror", that imply a specific causal relation. My guess is option (3): changes in technology (X) drive changes in the level of employment in various professions (E), they drive changes in sentiment towards various professions (S), and they drive changes in the frequency of various words for professions. But I have no rigorous evidence for this hypothesis.

Lines 8-10: "Cultivation theory suggests that prolonged exposure to the content we see on TV shapes our outlook and makes us believe that to be our reality [1]."

-- There should be some discussion of alternatives to Cultivation theory. For example: The Fruits of Cultivation Analysis: A Reexamination of Some Effects of Television Watching (pp. 287-302), Michael Hughes.

-- https://www.jstor.org/stable/2748103

-- Wikipedia says that there are two alternatives to Cultivation theory that are more widely used: "In a 2004 study, surveying almost 2,000 articles published in the top three mass communication journals since 1956, Jennings Bryant and Dorina Miron found that cultivation theory was the third most frequently utilized cultural theory."

-- https://en.wikipedia.org/wiki/Cultivation_theory

Lines 553-557: "The frequency trend of the SOC group does not reflect the frequency trend of all its professions. For example, Figs 8 e) and f) show increasing trends for many sales-related professions, but the overall frequency of the Sales Occupations SOC group decreased. A large proportion of Sales Occupations mentions are comprised of bankers and cashiers, whose frequencies have decreased."

-- Do the authors have an explanation for this? The paper should not be merely a collection of facts. Perhaps it is difficult to prove an explanation is correct, however it would be useful to suggest some hypotheses.

Lines 569-570: "Therefore, the number of positive sentiment mentions is greater than the number of negative sentiment mentions for all the SOC groups."

-- Again, some hypothetical explanations would be of interest to the reader.

Lines 589-598: "Fig 12 shows the Spearman's rank correlation coefficient [...] also one of the most positive sentiment trends."

-- Some hypothetical explanations would be of interest to the reader.

Lines 676-677: "In this section, we test the hypothesis that media stories reflect real-world events."

-- The word "reflect" here is somewhat metaphorical. There is an implication that real-world events cause media stories. But perhaps media stories cause real-world events. Or perhaps both are caused by some third factor. All that the authors can say with certainty is that there is correlation. I would very much welcome some explicit discussion of causality in this paper, but I don't think it's right to smuggle causality into the paper using terms such as "reflect". If the authors are unable to firmly state that there is a certain causal connection, they could still offer some tentative hypotheses about causation.

Lines 726-720: "Lastly, we observed that media frequency correlates with the employment trend of most SOC groups. Professions that employed more people were also more frequently mentioned in media content. This supports our hypothesis that media mirrors society and plays a role in our professional choices."

-- Again, the word "mirror" suggests that society is the cause and media is the effect. This paragraph starts with "correlates" (quite rightly) and ends with "mirrors" (somewhat unscientific). I encourage some discussion of causality, but it should be explicit and straightforward, acknowledging the difficulty of causal analysis.

Low-level Details

Abstract: "Societal ideas and trends dictate media narratives and cinematic depictions which in turn influences people's beliefs and perceptions of the real world."

-- "ideas", "trends", and "depictions" are plurals; therefore the verb should be "influence", not "influences".

Abstract: "study the effect of media attributes like genre, country of production, and title type"

-- replace "like" with "such as"

-- http://www.differencebetween.net/language/grammar-language/difference-between-such-as-and-like

Abstract: "and investigate if the incidence of professions in media subtitles correlate with their real-world employment statistics"

-- replace "if" with "whether"

-- https://www.merriam-webster.com/words-at-play/if-vs-whether-difference-usage

Lines 16-18: "Professions are paid, skilled work we perform to provide services to people and earn a livelihood. It defines our role in society and allows us to contribute to a nation's economy."

-- replace "It defines" with "They define"

Lines 21-22: "Therefore, it is essential to study and assess the changes in the occupational structure of the nation."

-- Why "of the nation"? Which nation are you talking about? Should it be "of a nation" or "of nations"?

Lines 40-41: "Several studies have examined the nature of the portrayal of different professions in popular media, like lawyers [13], accountants [14], physicians [15], and cops [16]."

-- replace "like" with "such as"

Lines 67-69: "2. We describe and share a new corpus of professional mentions, spanning 4,000 professions, 136,000 movies and TV shows, ranging over the years 1950 to 2017 68 (almost 7 decades) created by analyzing job title occurrences in media subtitles."

-- This would be a good place to add a reference to the list of references at the end of the paper. The reference would give a URL for downloading the new corpus. I think it is best to insert a reference here, rather than inserting a URL here.

Lines 72-75: "4. We analyze the frequency and contextual sentiment trend of professional mentions in media over time. We also investigate the presence of any correlation between incidence of professional mentions and the genre, title type, and country of production of the

movie or TV show."

-- This would be a good place to add references to the sentiment algorithm and the sentiment dataset.

Lines 105-108: "Kalisch et al. analyzed 670 nurse and 466 physician characters in novels, movies and television series, and concluded that compared to physicians, media nurses were consistently less central to the plot, less intelligent, rational, and less likely to exercise

clinical judgement."

-- I think "rational" should be "less rational".

Lines 108-111: "Smith et al. investigated gender representation of occupations in films, prime-time programs, and children TV shows, and found that females are grossly underrepresented compared to males in science, technology, engineering and math jobs (STEM) [24]."

-- The politically correct use of "female" and "male" is that these terms should only be used as adjectives ("female doctor", "male nurse"). The appropriate nouns are "men" and "women".

-- https://golin.com/2021/03/31/stop-using-female-when-you-mean-woman/

-- https://medium.com/fearless-she-wrote/woman-vs-female-67fd4c36fe59

-- https://www.buzzfeed.com/tracyclayton/stop-calling-women-females

Lines 130-131: "The TAC KBP track of entity discovery and linking introduced job titles as entity types to model the person:title relationship [30]."

-- What is TAC KBP? Add the long form in parentheses.

Lines 139-140: "Liu et al. combined dictionary lookups with semi-Markov CRF architectures and ..."

-- What is CRF? Add the long form in parentheses.

Lines 147-149: "Aside from careful human coding, such gazetteers can be constructed using different automated methods, including those that leverage existing knowledge bases like Wikipedia and WordNet [35, 36]."

-- replace "like" with "such as"

Lines 160-161: "Synsets are tagged by semantic classes (the complete tagset can be found at wordnet.princeton.edu)."

-- replace the URL with a proper reference at the end of the paper

Lines 218-220: "4. You're going to be a lousy architect. (Neutral) Explanation: The person towards which the negative sentiment is expressed, is not an architect."

-- "is not yet an architect"

Lines 221-222: "Benchmark ABSA datasets exist for several domains like question answering forums, customer reviews and tweets [56{58]."

-- replace "like" with "such as"

Lines 222-224: "Dong et al. proposed an adaptive recursive neural network for target-dependent twitter sentiment classification, that propagated the sentiments of words to target depending upon the context and syntactic relations [58]."

-- replace "words to target" with "words to their targets"

Lines 267-268: "An IMDB title can have multiple genres."

-- replace "IMDB" with "IMDb"

Lines 349-351: "We create a word-document search index using the Whoosh Python package for quick retrieval of mentions."

-- add a proper reference to Whoosh at the end of the paper, with a URL

Lines 360-361: "We apply the Stanford CoreNLP NER model [31] to find the named entity tag of words."

-- I think "tag" should be "tags"

Lines 381-382: "The CDM architecture masks less-SRD tokens, whereas the CDW architecture weighs them dynamically."

-- This needs more explanation. I don't know what it means.

Lines 740-741: "The profession taxonomy and sentiment-annotated subtitle corpus is publicly available at

" ext-link-type="uri" xlink:type="simple">https://github.com/usc-sail/mica-profession/tree/main/datasets."

-- Replace the URL with a proper reference at the end of the paper, including the URL in the reference, but not in the main body of the paper.

Reviewer #2: This paper has interesting results and methods, and, in my opinion, would be worthy of publication, after minor revisions.

Although the data and methodology are clearly presented, I would have appreciated the authors pointing out more explicitly their limitations.

The discussion section need substantive rework. While it summarizes well the main results of the article, it fails to engage in a reflective discussion with the findings of other studies. The discussion part would benefit from citing more social sciences references supporting the idea that "media mirrors society and plays a role in our professional choices". Also, it would be relevant to underline the main limitations of the study and to provide recommendations for future research in computational social sciences especially in other national and linguistic contexts

Overall, my recommendation is that the manuscript can be published with minor modifications

6. PLOS authors have the option to publish the peer review history of their article (what does this mean?). If published, this will include your full peer review and any attached files.

Reviewer #1: No

Reviewer #2: No

---

## [Author Response · Author response to Decision Letter 0]

19 Feb 2022

Dear Editor and Reviewers,

Thank you for your thoughtful feedback on our manuscript titled "Representation of professions in entertainment media: Insights into frequency and sentiment trends through computational text analysis". They have helped us to improve the paper significantly. Please find our response to your comments below.

The editor noted that grant information in the ‘Funding Information’ and ‘Financial Disclosures’ sections do not match. This is because our work was supported by a gift grant which did not have a grant number. We have added the U.S. Chamber of Commerce Foundation as a funding source without specifying a grant number in the revision. Our financial disclosure statement is: "The study was done at Signal Analysis and Interpretation Laboratory, University of Southern California, which is supported by a research award from the U.S. Chamber of Commerce Foundation (https://www.uschamberfoundation.org/). The funders had no role in study design, data collection and analysis, decision to publish, or preparation of work included in this submission."

Sabyasachee Baruah

Krishna Somandepalli

Shrikanth Narayanan

EDITOR

Comment 1 =============

Both reviewers agree that this is a worthwhile and carefully conducted study, something that I also sensed when reading the manuscript. All of us appreciate the clear writing style.

Response =============

Thank you.

Comment 2 =============

Reviewer 1 raises some "high-level" issues, mostly around the interpretation of your findings. They are particularly concerned about possible causal relationships between the correlates that you identify in your work, and I agree that this needs some finessing in a revision. As Reviewer 1 states, if one wishes to offer a particular causal interpretation, evidence needs to be provided to support this; otherwise, one should instead emphasize the nature of the correlations and at most tentatively suggest causal hypotheses that could be further explored in future work.

Response =============

We agree with the editor and reviewer 1, and acknowledge that we should refrain from causal assertions in our correlational study. We replace all potential ambiguous/ misleading statements like "media is a mirror to society" with "media and societal trends correlate with each other". While establishing causality would have been a much stronger result, showing that with our present experimental setting is difficult without accounting for additional demographic, political, technological, and cultural factors along with establishing the ecological validity of such findings given the constraints of the confounding variables. Our effort is a step towards this direction, and future work should include more controlled studies. We highlight this in our updated conclusion.

Revision =============

Abstract

Professions that employ more people showed increased media frequency.

Lines 684-686

In this section, we study the correlation between media frequency and real-world events, namely employment figures.

Lines 695-696

Therefore, the trend of media frequency correlates with the employment trend for most SOC groups.

Lines 760-763

Professions that employed more people were also more frequently mentioned in media content.

Therefore, media frequency correlates with employment. This finding however does not imply causality, a question which requires further systematic study.

Lines 790-791

Importantly, future work should also consider investigating causal relations, beyond correlations, between media representations and employment trends.

Comment 3 =============

At the same time, Reviewer 1 notes that the manuscript presents a body of facts, but does not offer possible explanations for them - so clearly some speculation is to be appreciated, as long as it is presented as such.

Response =============

We have reworked the discussion section to suggest possible reasons behind the observed media frequency and sentiment trends. We could not always find proper references to support our hypotheses, and sometimes even found published studies contradicting our media trends. We have included the relevant published work and have tried our best to explain the media trends.

Revision =============

Lines 703-706

We observed that gender-neutral terms like massage therapists and flight-attendants are becoming more frequent than their gendered counterparts. This might be due to the increased conversations around using gender-neutral terms among the youth and the LGBTQ+ community [75, 76].

Lines 707-712

The frequency of some female job titles like waitresses, congresswomen, and policewomen has either increased or remained steady relative to the corresponding male job titles -- waiters, congressmen, and policemen -- which have decreased in frequency.

The increasing representation of women in police and government might be because of improvement in equal opportunity legislation, women's rights movements [77], and constitutional amendments for women's suffrage [78].

Lines 714-718

The frequency of STEM, sports, arts, and design occupations has increased, whereas the frequency of construction, farming, and manual labor jobs has decreased.

Improvement in cinematography and visual effects technology, space travel, the emergence of the world wide web, and sports documentaries might have contributed to the increased fascination of science and sports in media [79, 80].

We have similarly revised Lines 719-748 and offered possible hypotheses to explain the observed media trends.

REVIEWER 1

Comment 1 =============

Abstract: "Professions that employ more people have increased media frequency, supporting our hypothesis that media acts as a mirror to society." -- Let E be the levels of employment in various professions. Let F be the frequency of various words for professions. Let S be the sentiment of various words applied to professions. The sentence quoted here suggests that variation in E causes variation in F and S: that is, E is the real thing, whereas F and S are merely reflections (mirrors) of E. The paper makes it clear that E is correlated with F and S, but the paper does not demonstrate how the causal connection goes. There are (at least) three possibilities: (1) Variation in E causes variations in F and S. (2) Variations in F and S cause variations in E. (3) Variation in some other thing, X, causes variation in E, F, and S. All three of these causal connections seem possible to me, and it is not at all clear which is the actual connection. The authors need to either (A) be clear that they have only demonstrated correlation and not causation or (B) provide solid evidence that there is indeed a specific causal relation connecting E, F, and S. If the authors choose (A) (causation unknown), then they should remove all expressions, such as "acts as a mirror", that imply a specific causal relation. My guess is option (3): changes in technology (X) drive changes in the level of employment in various professions (E), they drive changes in sentiment towards various professions (S), and they drive changes in the frequency of various words for professions. But I have no rigorous evidence for this hypothesis.

Lines 676-677: "In this section, we test the hypothesis that media stories reflect real-world events."

-- The word "reflect" here is somewhat metaphorical. There is an implication that real-world events cause media stories. But perhaps media stories cause real-world events. Or perhaps both are caused by some third factor. All that the authors can say with certainty is that there is correlation. I would very much welcome some explicit discussion of causality in this paper, but I don't think it's right to smuggle causality into the paper using terms such as "reflect". If the authors are unable to firmly state that there is a certain causal connection, they could still offer some tentative hypotheses about causation.

Lines 726-720: "Lastly, we observed that media frequency correlates with the employment trend of most SOC groups. Professions that employed more people were also more frequently mentioned in media content. This supports our hypothesis that media mirrors society and plays a role in our professional choices."

-- Again, the word "mirror" suggests that society is the cause and media is the effect. This paragraph starts with "correlates" (quite rightly) and ends with "mirrors" (somewhat unscientific). I encourage some discussion of causality, but it should be explicit and straightforward, acknowledging the difficulty of causal analysis.

Response =============

We agree that correlation between E and F (or S) does not imply causation. It was not our intention to suggest so either. Therefore, we have rephrased all statements relating media trends and employment to explicitly contain the word “correlation”, and removed misleading statements like “media is a mirror to society”.

Revision =============

Same as the revision to the Editor's second comment (see above).

Comment 2 =============

Lines 8-10: "Cultivation theory suggests that prolonged exposure to the content we see on TV shapes our outlook and makes us believe that to be our reality [1]."

-- There should be some discussion of alternatives to Cultivation theory. For example: The Fruits of Cultivation Analysis: A Reexamination of Some Effects of Television Watching (pp. 287-302), Michael Hughes.

-- https://www.jstor.org/stable/2748103

-- Wikipedia says that there are two alternatives to Cultivation theory that are more widely used: "In a 2004 study, surveying almost 2,000 articles published in the top three mass communication journals since 1956, Jennings Bryant and Dorina Miron found that cultivation theory was the third most frequently utilized cultural theory."

-- https://en.wikipedia.org/wiki/Cultivation_theory

Response =============

We have provided some discussion on alternatives to Cultivation theory. This debate surrounding the relationship between television viewing and human behavior and beliefs suggests that we require better designed and more demographically inclusive surveys. We have added the suggested citations for completeness.

Revision =============

Lines 8 - 15

Cultivation theory suggests that prolonged exposure to the content we see on TV shapes our outlook and makes us believe that to be our reality [1]. Several social studies have explored the application of cultivation theory and confirmed its validity [2–4]. However, there are social scientists that have also questioned the validity of the cultivation theory because it ignores socioeconomic factors [5], living conditions [6], and differences in the portrayal of violence on television [7]. Still, this discussion about television and the cultivation of beliefs inspires us to examine their relationship.

Comment 3 =============

Lines 553-557: "The frequency trend of the SOC group does not reflect the frequency trend of all its professions. For example, Figs 8 e) and f) show increasing trends for many sales-related professions, but the overall frequency of the Sales Occupations SOC group decreased. A large proportion of Sales Occupations mentions are comprised of bankers and

cashiers, whose frequencies have decreased."

-- Do the authors have an explanation for this? The paper should not be merely a collection of facts. Perhaps it is difficult to prove an explanation is correct, however it would be useful to suggest some hypotheses.

Lines 569-570: "Therefore, the number of positive sentiment mentions is greater than the number of negative sentiment mentions for all the SOC groups."

-- Again, some hypothetical explanations would be of interest to the reader.

Lines 589-598: "Fig 12 shows the Spearman's rank correlation coefficient [...] also one of the most positive sentiment trends."

-- Some hypothetical explanations would be of interest to the reader.

Response =============

Similar to the response to the Editor’s third comment, we have included possible hypotheses with relevant citations to explain the observed frequency and sentiment media trends. The analysis section contains the facts and the discussion section provides possible reasons for these facts.

Revision =============

Same as the revision to the Editor’s third comment.

We summarize our edits for the low level comments raised by reviewer 1 here:

1. We replace “like” with “such as” at the suggested places.

2. We replace “if” with “whether” at the suggested places.

3. We replace URLs with proper citations for the profession corpus, sentiment dataset, and sentiment algorithm.

4. We replace “males” and “females” with “men” and “women” respectively.

5. We expand the abbreviated forms of CRF and TAC KBP.

6. We describe the LCF BERT model in simpler terms.

Lines 386-391

The LCF model defines a value called semantic relative distance (SRD) for each word in the sentence. SRD is the absolute difference between the word position and the target (professional mention) position in the sentence. The model masks or weighs down the output features of words whose SRD is larger than some threshold, that is, the model reduces the effect of words that are farther away from the profession word in determining the sentiment.

REVIEWER 2

Comment 1 =============

Although the data and methodology are clearly presented, I would have appreciated the authors pointing out more explicitly their limitations….Also, it would be relevant to underline the main limitations of the study and to provide recommendations for future research in computational social sciences especially in other national and linguistic contexts.

Response =============

We acknowledge that the earlier version of the manuscript lacked a discussion of the limitations of our work. While ours is the first large-scale computational analysis on the representation of professions in media subtitles, we can further improve our methods through more sophisticated modeling. Moreover, our domain choice limits the information we can glean about the media portrayal of professions. We have described these limitations and more at the end of the discussion section. 

Revision =============

Lines 764-779

There are several limitations of the present work. First, subtitles provide only a limited view of the complete movie or TV show. Features, such as the character's behavior on screen, their interactions with other people, and their character arc can only be studied by carefully viewing the entire movie or TV episode. Therefore, our representation measures of frequency and sentiment of job titles in subtitles do not capture all aspects of their media portrayal.

Second, our frequency and sentiment analysis does not control for the frequency or sentiment of the professional words in everyday language. For example, the frequency of policewomen might have been less than the frequency of policemen in media subtitles because that is how their usage evolved in literature. A solution to this might be to use the corresponding Google ngram frequencies [23] of the professional words as covariates in our analysis. Third, our study has primarily focused on the representation of professions in the US and the UK (see the distribution of production country in section \\nameref{subsec:opensubtitles}). We need to adapt our taxonomy and models to work across different cultures and languages. Our profession taxonomy might not contain job titles that are prevalent in other countries, and we will have to add the professions specific to a local region for a comprehensive study.

Comment 2 =============

The discussion section need substantive rework. While it summarizes well the main results of the article, it fails to engage in a reflective discussion with the findings of other studies. The discussion part would benefit from citing more social sciences references supporting the idea that "media mirrors society and plays a role in our professional choices".

Response =============

We have reworked the discussion section extensively and provided possible reasons behind the observed media frequency and sentiment trends of professions. 

Revision =============

Please see the revision to the Editor’s third comment.

REFERENCES

We have not removed any previous references from our manuscript. We have cited the following new references:

5. Hughes M. The Fruits of Cultivation Analysis: A Reexamination of Some Effects

of Television Watching. The Public Opinion Quarterly. 1980;44(3):287–302.

6. Chandler D. Cultivation Theory; 1995.

http://visual-memory.co.uk/daniel//Documents/short/cultiv.html.

7. Newcomb H. Assessing the Violence Profile Studies of Gerbner and Gross: A

Humanistic Critique and Suggestion. Communication Research.

1978;5(3):264–282. doi:10.1177/009365027800500303.

26. Baruah S, Somandepalli K, Narayanan S. GitHub repository: Representation of

professions in entertainment media; 2022. Available from:

https://github.com/sabyasachee/mica-profession.

44. Princeton University. WordNet lexicographer file names and numbers; 2010.

https://wordnet.princeton.edu/documentation/lexnames5wn.

73. Chaput M. GitHub repository: Whoosh; 2018. Available from:

https://github.com/mchaput/whoosh.

75. Berger M. A guide to how gender-neutral language is developing around the

world; 2019. https://www.washingtonpost.com/world/2019/12/15/

guide-how-gender-neutral-language-is-developing-around-world/.

76. Williams D. The Rise of Gender Neutrality and its Impact on Language; 2018. https://www.translatemedia.com/us/blog-usa/rise-gender-neutrality-impact-language/

77. Nicholson P. The second wave: A reader in feminist theory. Psychology Press;

1997.

78. Brown JK. The Nineteenth Amendment and women’s equality. The Yale Law

Journal. 1993;102(8):2175–2204.

79. Geraghty L. American science fiction film and television. Berg; 2009.

80. Jenns N. Why Suddenly Real-Life Sports Documentaries Trend in Film Industry?;

2021. https://www.tetongravity.com/story/news/ why-suddenly-real-life-sports-documentaries-trend-in-film-industry.

81. Hauer KE, Durning SJ, Kernan WN, Fagan MJ, Mintz M, O’Sullivan PS, et al.

Factors Associated With Medical Students’ Career Choices Regarding Internal

Medicine. JAMA. 2008;300(10):1154–1164. doi:10.1001/jama.300.10.1154.

82. Dalen JE, Ryan KJ, Alpert JS. Where have the generalists gone? They became

specialists, then subspecialists. The American journal of medicine.

2017;130(7):766–768.

83. Reimink T. Are Bank Tellers and Retail Cashiers Experiencing Similar

Transformations?; 2017. https://www.linkedin.com/pulse/

bank-tellers-retail-cashiers-experiencing-similar-timothy-reimink/.

84. Wong JC. End of the checkout line: the looming crisis for American cashiers;

2017. https://www.theguardian.com/technology/2017/aug/16/

retail-industry-cashier-jobs-technology-unemployment.

85. Funk C, Hefferon M, Kennedy B, Johnson C. Trust and mistrust in Americans’

views of scientific experts. Pew Research Center. 2019;2:1–96.

86. Doherty C, Kiley J, Daniller A, Jones B, Hartig H, Dunn A, et al.. Majority of

public favors giving civilians the power to sue police officers for misconduct; 2020.

87. Flanagan TJ, Vaughn MS. Public opinion about police abuse of force. Police

violence: Understanding and controlling police abuse of force. 1996;30(5):397–408.

88. Kennedy BR, Mathis CC, Woods AK. African Americans and their distrust of

the health care system: healthcare for diverse populations. Journal of cultural

diversity. 2007;14(2).

---

## [Editor Report · Decision Letter 1]

7 Mar 2022

PONE-D-21-27904R1Representation of professions in entertainment media: Insights into frequency and sentiment trends through computational text analysisPLOS ONE

Dear Dr. Baruah,

Thank you for resubmitting your manuscript to PLOS ONE. I have read through your response and the revisions to the paper. As stated in my previous message, my preference in general is to avoid a second round of review and I think in this instance this would be achievable if you were able to make some further small revisions to the Discussion section.In response to me and Reviewer 1, you have included some suggestions as to factors that might be responsible for the changes that have been detected in the corpus. Clearly these are speculative, as we suggested they might be, but I think some further text is necessary to draw attention to the speculative nature of the proposed connections, and the need for further work to establish whether or not they actually hold.My suggestion would be to insert some text at the start of the Discussion section, nothing that you will tentatively suggest some possible causes of the changes. This might allow you to deviate a little from the "X might be because of Y" template you followed throughout the discussion, which I found a little repetitive. I think it would be more satisfactory to employ forms of language like "There is some evidence of Y [reference], which could lead to [X happening]" and variants that remove repetition. Where possible it could be useful to add comments as to the mechanism by which it might filter through to your dataset and/or what you might need to do in practice to demonstrate the connection. I think overall this would make for a more satisfying and professional discussion, and if you were able to achieve this I would expect to be able to accept the manuscript without the need to consult the reviewers again.I would hope that this would not take you more than a few hours.

We look forward to receiving your revised manuscript.

Kind regards,

Richard A Blythe

Academic Editor

PLOS ONE
---

## [Author Response · Author response to Decision Letter 1]

11 Apr 2022

Dear Editor and Reviewers,

Thank you for your thoughtful feedback on our manuscript titled "Representation of professions in entertainment media: Insights into frequency and sentiment trends through computational text analysis". They have helped us to improve the paper significantly. Please find our response to your comments below.

The editor noted that grant information in the ‘Funding Information’ and ‘Financial Disclosures’ sections do not match. This is because our work was supported by a gift grant which did not have a grant number. We have added the U.S. Chamber of Commerce Foundation as a funding source without specifying a grant number in the revision. Our financial disclosure statement is: "The study was done at Signal Analysis and Interpretation Laboratory, University of Southern California, which is supported by a research award from the U.S. Chamber of Commerce Foundation (https://www.uschamberfoundation.org/). The funders had no role in study design, data collection and analysis, decision to publish, or preparation of work included in this submission." As per the editor’s suggestion, we have included the updated financial disclosure statement in the cover letter as well.

We have also included a section below called “Editor(Email)” below to respond to the paraphrasing suggestions raised by the editor in the email sent on Mar 7th with the subject: PLOS ONE Decision: Revision required [PONE-D-21-27904R1] - [EMID:931db3d1d60343a8].

Sabyasachee Baruah

Krishna Somandepalli

Shrikanth Narayanan

Editor (Email)

Comment

In response to me and Reviewer 1, you have included some suggestions as to factors that might be responsible for the changes that have been detected in the corpus. Clearly these are speculative, as we suggested they might be, but I think some further text is necessary to draw attention to the speculative nature of the proposed connections, and the need for further work to establish whether or not they actually hold.

My suggestion would be to insert some text at the start of the Discussion section, noting that you will tentatively suggest some possible causes of the changes. This might allow you to deviate a little from the "X might be because of Y" template you followed throughout the discussion, which I found a little repetitive. I think it would be more satisfactory to employ forms of language like "There is some evidence of Y [reference], which could lead to [X happening]" and variants that remove repetition. Where possible it could be useful to add comments as to the mechanism by which it might filter through to your dataset and/or what you might need to do in practice to demonstrate the connection.

Response

We changed the introductory passage in the discussion section to emphasize that we selected a few interesting trends to discuss and our hypotheses to explain these trends are purely speculative. We also state that verifying our hypotheses would require further research and controlled experiments. As of now we are still unsure what kind of future experiments we would need because it is difficult to design studies that can establish any form of relationship between media narratives and real-world societal factors. Nevertheless, we suggest some ideas in our discussion. 

We have also created a visualization tool that allows users to explore the trends in our subtitle corpus. We cite it in the discussion section. Also, as per the editor’s paraphrasing-related requests, we have varied the language to deviate from the “X might be because of Y” template.

Revision

Lines 704-715

The frequency and sentiment expressed towards professionals vary in media content

depending on the character archetype, the story being told, year of production, and 

genre, among other variables. We developed a visualization tool [75] to examine the 

frequency, sentiment, media attribute, and employment trends in our subtitle corpus. 

While examining the wealth of results using this tool, a few general findings and specific 

trends stood out to us. In this section, we highlight these trends, discuss some reasons 

that may be suggestive for explaining the observed results, and try to reconcile these 

findings with those in relevant studies, both from the real world and the media domain. 

We underscore that our interpretation is purely speculative and requires careful, 

controlled experiments and surveys to further validate the claims. Also, this discussion 

is by no means an effort to explain all our findings, and we hope future research in this 

space may further explain some of the results we found in this large-scale analysis.

Lines 718-720

We suggest that this trend is due to the increased awareness stemming from conversations about gender-neutral terms among the youth, parents, and the LGBTQ+ community [76, 77]

Lines 734-736

These advances and interests potentially explain the increase in the frequency of STEM-related and sports portrayals on the big screen.

Lines 741-751

We posit that as new and specialized professions become popular career choices and essential 

in catering to the needs of the continuously evolving society, they also attract the interest of media creators. 

Technology improvements in computerized financial transactions (barcode scanning, automatic teller machines, and mobile banking) 

and customers’ preference for self-service checkouts probably contributed to the 

decrease in employment of bankers and cashiers [84, 85], or at least their explicit mentions in media narratives.

Lines 773-781

We have made several hypotheses to explain the observed frequency and sentiment

trends and require controlled experiments to verify these claims. It is challenging to

design such studies because we are trying to relate media narratives to real-world events

– two very different domains. As of now, we are unsure regarding what types of

experiments are required, but we suggest some ideas. We can survey scriptwriters and

movie directors to find what societal events inspired their stories and creative decisions.

We can also try to map change points in media trends to real-world events to explain

their possible cascading effects on media narratives and genres. We leave this task for

future research.

Editor

Comment 1

Both reviewers agree that this is a worthwhile and carefully conducted study, something that I also sensed when reading the manuscript. All of us appreciate the clear writing style.

Response

Thank you.

Comment 2

Reviewer 1 raises some "high-level" issues, mostly around the interpretation of your findings. They are particularly concerned about possible causal relationships between the correlates that you identify in your work, and I agree that this needs some finessing in a revision. As Reviewer 1 states, if one wishes to offer a particular causal interpretation, evidence needs to be provided to support this; otherwise, one should instead emphasize the nature of the correlations and at most tentatively suggest causal hypotheses that could be further explored in future work.

Response

We agree with the editor and reviewer 1, and acknowledge that we should refrain from causal assertions in our correlational study. We replace all potential ambiguous/ misleading statements like "media is a mirror to society" with "media and societal trends correlate with each other". While establishing causality would have been a much stronger result, showing that with our present experimental setting is difficult without accounting for additional demographic, political, technological, and cultural factors along with establishing the ecological validity of such findings given the constraints of the confounding variables. Our effort is a step towards this direction, and future work should include more controlled studies. We highlight this in our updated conclusion and discussion sections.

Revision

Abstract

Professions that employ more people have showed increased media frequency, supporting our hypothesis that media acts as a mirror to society.

Lines 686-688

In this section, we test the hypothesis that media stories reflect real-world events study the correlation between media frequency and real-world events, namely employment figures.

Lines 697-698

Therefore, the trend of media frequency mirrors correlates with the employment trend for most SOC groups.

Lines 798-802

We observed a significant positive correlation between media frequency and employment trends of most SOC groups. Professions that typically employ more people were also more frequently mentioned in media content. We note that these correlation findings do not imply causality; a question that requires further systematic study.

Lines 831-832

Importantly, future work should also consider investigating causal relations, beyond correlations, between media representations and employment trends.

Comment 3

At the same time, Reviewer 1 notes that the manuscript presents a body of facts, but does not offer possible explanations for them - so clearly some speculation is to be appreciated, as long as it is presented as such.

Response

We have reworked the discussion section to suggest possible reasons behind the observed media frequency and sentiment trends. We could not always find proper references to support our hypotheses, and sometimes even found published studies contradicting our media trends. We have included the relevant published work and have tried our best to explain the media trends.

Revision

Lines 717-720

We observed that gender-neutral terms like massage therapists and flight-attendants are

becoming more frequent than their gendered counterparts. We suggest that this trend is

due to the increased awareness stemming from conversations about gender-neutral

terms among the youth, parents, and the LGBTQ+ community [76, 77].

Lines 720-727

Analyzing explicit gendered profession terms, we observed that the frequency of some female job titles such as waitresses, congresswomen, and policewomen has either increased or

remained steady relative to the corresponding male job titles (i.e., waiters, congressmen,

and policemen). Equal opportunity legislation, women’s rights movements [78], and

constitutional amendments for women’s suffrage [79] have improved the representation

of women in police and government. This may have driven a similar trend of increasing

female representation in media content

Lines 730-736

The frequency of STEM, sports, arts, and design occupations has increased, whereas

the frequency of construction, farming, and manual labor jobs has decreased.

Improvements in cinematography and visual effects technology, space travel, the

emergence of the world wide web, and sports documentaries have contributed to the

increased fascination of science and sports in media [80, 81]. These advances and

interests potentially explain the increase in the frequency of STEM-related and sports

portrayals on the big screen.

We have similarly revised the entire discussion section and offered possible hypotheses to explain the observed media trends.

Reviewer 1

Comment 1

Abstract: "Professions that employ more people have increased media frequency, supporting our hypothesis that media acts as a mirror to society." -- Let E be the levels of employment in various professions. Let F be the frequency of various words for professions. Let S be the sentiment of various words applied to professions. The sentence quoted here suggests that variation in E causes variation in F and S: that is, E is the real thing, whereas F and S are merely reflections (mirrors) of E. The paper makes it clear that E is correlated with F and S, but the paper does not demonstrate how the causal connection goes. There are (at least) three possibilities: (1) Variation in E causes variations in F and S. (2) Variations in F and S cause variations in E. (3) Variation in some other thing, X, causes variation in E, F, and S. All three of these causal connections seem possible to me, and it is not at all clear which is the actual connection. The authors need to either (A) be clear that they have only demonstrated correlation and not causation or (B) provide solid evidence that there is indeed a specific causal relation connecting E, F, and S. If the authors choose (A) (causation unknown), then they should remove all expressions, such as "acts as a mirror", that imply a specific causal relation. My guess is option (3): changes in technology (X) drive changes in the level of employment in various professions (E), they drive changes in sentiment towards various professions (S), and they drive changes in the frequency of various words for professions. But I have no rigorous evidence for this hypothesis.

Lines 676-677: "In this section, we test the hypothesis that media stories reflect real-world events."

-- The word "reflect" here is somewhat metaphorical. There is an implication that real-world events cause media stories. But perhaps media stories cause real-world events. Or perhaps both are caused by some third factor. All that the authors can say with certainty is that there is correlation. I would very much welcome some explicit discussion of causality in this paper, but I don't think it's right to smuggle causality into the paper using terms such as "reflect". If the authors are unable to firmly state that there is a certain causal connection, they could still offer some tentative hypotheses about causation.

Lines 726-720: "Lastly, we observed that media frequency correlates with the employment trend of most SOC groups. Professions that employed more people were also more frequently mentioned in media content. This supports our hypothesis that media mirrors society and plays a role in our professional choices."

-- Again, the word "mirror" suggests that society is the cause and media is the effect. This paragraph starts with "correlates" (quite rightly) and ends with "mirrors" (somewhat unscientific). I encourage some discussion of causality, but it should be explicit and straightforward, acknowledging the difficulty of causal analysis.

Response

We agree that correlation between E and F (or S) does not imply causation. It was not our intention to suggest so either. Therefore, we have rephrased all statements relating media trends and employment to explicitly contain the word “correlation”, and removed misleading statements like “media is a mirror to society”.

Revision

Same as the revision to the Editor's second comment (see above).

Comment 2

Lines 8-10: "Cultivation theory suggests that prolonged exposure to the content we see on TV shapes our outlook and makes us believe that to be our reality [1]."

-- There should be some discussion of alternatives to Cultivation theory. For example: The Fruits of Cultivation Analysis: A Reexamination of Some Effects of Television Watching (pp. 287-302), Michael Hughes.

-- https://www.jstor.org/stable/2748103

-- Wikipedia says that there are two alternatives to Cultivation theory that are more widely used: "In a 2004 study, surveying almost 2,000 articles published in the top three mass communication journals since 1956, Jennings Bryant and Dorina Miron found that cultivation theory was the third most frequently utilized cultural theory."

-- https://en.wikipedia.org/wiki/Cultivation_theory

Response

We have provided some discussion on alternatives to Cultivation theory. This debate surrounding the relationship between television viewing and human behavior and beliefs suggests that we require better designed and more demographically inclusive surveys. We have added the suggested citations for completeness.

Revision

Lines 8 - 15

 Cultivation theory suggests that prolonged exposure to the content we see on TV shapes our 

outlook and makes us believe that to be our reality [1]. Several social studies have 

explored the application of cultivation theory and confirmed its validity [2–4]. However, 

there are social scientists that have also questioned the validity of the cultivation theory 

because it ignores socioeconomic factors [5], living conditions [6], and differences in the 

portrayal of violence on television [7]. Still, this discussion about television and the 

cultivation of beliefs inspires us to examine their relationship

Comment 3

Lines 553-557: "The frequency trend of the SOC group does not reflect the frequency trend of all its professions. For example, Figs 8 e) and f) show increasing trends for many sales-related professions, but the overall frequency of the Sales Occupations SOC group decreased. A large proportion of Sales Occupations mentions are comprised of bankers and

cashiers, whose frequencies have decreased."

-- Do the authors have an explanation for this? The paper should not be merely a collection of facts. Perhaps it is difficult to prove an explanation is correct, however it would be useful to suggest some hypotheses.

Lines 569-570: "Therefore, the number of positive sentiment mentions is greater than the number of negative sentiment mentions for all the SOC groups."

-- Again, some hypothetical explanations would be of interest to the reader.

Lines 589-598: "Fig 12 shows the Spearman's rank correlation coefficient [...] also one of the most positive sentiment trends."

-- Some hypothetical explanations would be of interest to the reader.

Response

Similar to the response to the Editor’s third comment, we have included possible hypotheses with relevant citations to explain the observed frequency and sentiment media trends. The analysis section contains the facts and the discussion section provides possible reasons for these facts.

Revision

Same as the revision to the Editor’s third comment.

We summarize our edits for the low level comments raised by reviewer 1 here:

We replace “like” with “such as” at the suggested places.

We replace “if” with “whether” at the suggested places.

We replace URLs with proper citations for the profession corpus, sentiment dataset, and sentiment algorithm.

We replace “males” and “females” with “men” and “women” respectively.

We expand the abbreviated forms of CRF and TAC KBP.

We describe the LCF BERT model in simpler terms.

Lines 386-391

The LCF model defines a value called semantic relative distance (SRD) for each word in the sentence. SRD is the absolute difference between the word position and the target (professional mention) position in the sentence. The model masks or weighs down the output features of words whose SRD is larger than some threshold, that is, the model reduces the effect of words that are farther away from the profession word in determining the sentiment.

Reviewer 2

Comment 1

Although the data and methodology are clearly presented, I would have appreciated the authors pointing out more explicitly their limitations….Also, it would be relevant to underline the main limitations of the study and to provide recommendations for future research in computational social sciences especially in other national and linguistic contexts.

Response

We acknowledge that the earlier version of the manuscript lacked a discussion of the limitations of our work. While ours is the first large-scale computational analysis on the representation of professions in media subtitles, we can further improve our methods through more sophisticated modeling. Moreover, our domain choice limits the information we can glean about the media portrayal of professions. We have described these limitations and more at the end of the discussion section. 

Revision

Lines 803-820

There are several limitations of the present work. First, subtitles provide only a limited 

view of the complete movie or TV show. Features, such as the character’s behavior on 

screen, their interactions with other people, and their character arc can only be studied 

by carefully viewing the entire movie or TV episode. Therefore, our representation 

measures of frequency and sentiment of job titles in subtitles do not capture all aspects 

of their media portrayal. Second, our frequency and sentiment analyses do not control 

for the frequency or sentiment expressed toward these professions in everyday language. 

For example, the frequency of policewomen might have been less than the frequency of 

policemen in media subtitles because that is how their usage evolved in literature. A 

possible solution is to use the corresponding Google ngram frequencies [23] of the 

professional words as covariates in our analysis. Third, our study has primarily focused 

on the representation of professions in the US and the UK (see the distribution of 

production country in section OpenSubtitles). One needs to adapt our proposed 

taxonomy and models for different cultures and languages. Our profession taxonomy

does not contain all job titles in other countries. Depending on the application of this 

taxonomy, one may have to add professions specific to a local region for a 

comprehensive study. 

Comment 2

The discussion section need substantive rework. While it summarizes well the main results of the article, it fails to engage in a reflective discussion with the findings of other studies. The discussion part would benefit from citing more social sciences references supporting the idea that "media mirrors society and plays a role in our professional choices".

Response

We have reworked the discussion section extensively and provided possible reasons behind the observed media frequency and sentiment trends of professions. 

Revision

Please see the revision to the Editor’s third comment.

References

We have not removed any previous references from our manuscript. We have cited the following new references:

5. Hughes M. The Fruits of Cultivation Analysis: A Reexamination of Some Effects

of Television Watching. The Public Opinion Quarterly. 1980;44(3):287–302.

6. Chandler D. Cultivation Theory; 1995.

http://visual-memory.co.uk/daniel//Documents/short/cultiv.html.

7. Newcomb H. Assessing the Violence Profile Studies of Gerbner and Gross: A

Humanistic Critique and Suggestion. Communication Research.

1978;5(3):264–282. doi:10.1177/009365027800500303.

26. Baruah S, Somandepalli K, Narayanan S. GitHub repository: Representation of

professions in entertainment media; 2022. Available from:

https://github.com/sabyasachee/mica-profession.

44. Princeton University. WordNet lexicographer file names and numbers; 2010.

https://wordnet.princeton.edu/documentation/lexnames5wn.

75. Chaput M. GitHub repository: Whoosh; 2018. Available from:

https://github.com/mchaput/whoosh.

78. Berger M. A guide to how gender-neutral language is developing around the

world; 2019. https://www.washingtonpost.com/world/2019/12/15/

guide-how-gender-neutral-language-is-developing-around-world/.

79. Williams D. The Rise of Gender Neutrality and its Impact on Language; 2018. https://www.translatemedia.com/us/blog-usa/rise-gender-neutrality-impact-language/

80. Nicholson P. The second wave: A reader in feminist theory. Psychology Press;

1997.

81. Brown JK. The Nineteenth Amendment and women’s equality. The Yale Law

Journal. 1993;102(8):2175–2204.

82. Geraghty L. American science fiction film and television. Berg; 2009.

83. Jenns N. Why Suddenly Real-Life Sports Documentaries Trend in Film Industry?;

2021. https://www.tetongravity.com/story/news/ why-suddenly-real-life-sports-documentaries-trend-in-film-industry.

84. Hauer KE, Durning SJ, Kernan WN, Fagan MJ, Mintz M, O’Sullivan PS, et al.

Factors Associated With Medical Students’ Career Choices Regarding Internal

Medicine. JAMA. 2008;300(10):1154–1164. doi:10.1001/jama.300.10.1154.

85. Dalen JE, Ryan KJ, Alpert JS. Where have the generalists gone? They became

specialists, then subspecialists. The American journal of medicine.

2017;130(7):766–768.

86. Reimink T. Are Bank Tellers and Retail Cashiers Experiencing Similar

Transformations?; 2017. https://www.linkedin.com/pulse/

bank-tellers-retail-cashiers-experiencing-similar-timothy-reimink/.

87. Wong JC. End of the checkout line: the looming crisis for American cashiers;

2017. https://www.theguardian.com/technology/2017/aug/16/

Retail-industry-cashier-jobs-technology-unemployment.

88. Flanagan TJ, Vaughn MS. Public opinion about police abuse of force. Police

violence: Understanding and controlling police abuse of force. 1996;30(5):397–408.

89. Funk C, Hefferon M, Kennedy B, Johnson C. Trust and mistrust in Americans’

views of scientific experts. Pew Research Center. 2019;2:1–96.

90. Doherty C, Kiley J, Daniller A, Jones B, Hartig H, Dunn A, et al.. Majority of

public favors giving civilians the power to sue police officers for misconduct; 2020.

91. Kennedy BR, Mathis CC, Woods AK. African Americans and their distrust of

the health care system: healthcare for diverse populations. Journal of cultural

diversity. 2007;14(2).

92. Ramakrishna A, Martinez VR, Malandrakis N, Singla K, Narayanan S. Linguistic

Analysis of differences in portrayal of movie characters. In: Proceedings of the 55th Annual Meeting of the Association for Computational Linguistics (Volume 1: Long Papers). Vancouver, Canada: Association for Computational Linguistics; 2017. p. 1669–1678. Available from: https://aclanthology.org/P17-1153.

---

## [Editor Report · Decision Letter 2]

18 Apr 2022

Representation of professions in entertainment media: Insights into frequency and sentiment trends through computational text analysis

PONE-D-21-27904R2

Dear Dr. Baruah,

Thank you for reconsidering the discussion section of this manuscript, which is now much improved. I'm pleased to inform you that your manuscript has been judged scientifically suitable for publication and will be formally accepted for publication once it meets all outstanding technical requirements.

Kind regards,

Richard A Blythe

Academic Editor

PLOS ONE
---

## [Editor Report · Acceptance letter]

21 Apr 2022

PONE-D-21-27904R2 

Representation of professions in entertainment media: Insights into frequency and sentiment trends through computational text analysis 

Dear Dr. Baruah:

I'm pleased to inform you that your manuscript has been deemed suitable for publication in PLOS ONE. Congratulations! Your manuscript is now with our production department. 

Kind regards, 

on behalf of

Prof. Richard A Blythe 

Academic Editor

PLOS ONE